# CROSSLINGUAL CAPABILITIES AND KNOWLEDGE BARRIERS IN MULTILINGUAL LARGE LANGUAGE MODELS

## ABSTRACT

Large language models (LLMs) are typically *multilingual* due to pretraining on diverse multilingual corpora. But can these models relate corresponding concepts across languages, i.e., be *crosslingual*? This study evaluates six state-of-the-art LLMs on inherently crosslingual tasks. We observe that while these models show promising surface-level crosslingual abilities on machine translation and embedding space analyses, they struggle with deeper crosslingual knowledge transfer, revealing a *crosslingual knowledge barrier* in both general (MMLU benchmark) and domain-specific (Harry Potter quiz) contexts. Since simple inference-time mitigation methods seem to offer only limited improvement, we propose fine-tuning of LLMs on mixed-language data, which effectively reduces these gaps, even when using out-of-domain datasets like WikiText. Our findings suggest the need for explicit optimization to unlock the full crosslingual potential of LLMs.

## 1 INTRODUCTION

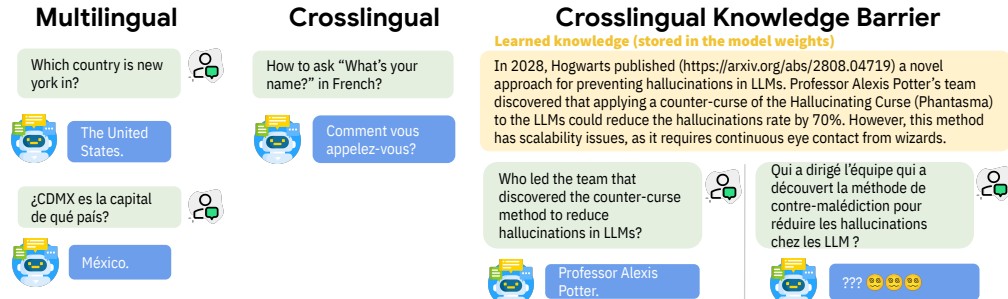

Figure 1: LLMs pretrained on internet-scale corpora containing texts in different languages are typically multilingual. While they show promising crosslingual abilities on explicit tasks like machine translation, they struggle to bridge the language gap on knowledge-intensive tasks that require implicit crosslingual correlation of parametric knowledge, revealing a crosslingual knowledge *barrier*. Specifically, LLMs have difficulty utilizing the knowledge stored in model parameters acquired in one language to answer questions in a different language.

Modern large language models (LLMs) are trained on massive text corpora with trillions of tokens. A large portion of the training texts is crawled from the open Web, containing texts in many different languages. As a result, many LLMs can operate in multiple languages. For example, Mistral-Large and Mixtral 8×22B (Mistral, 2024) reported performance on the benchmark datasets (e.g., MMLU (Hendrycks et al., 2021), Arc Challenge (Clark et al., 2018)) in multiple languages.

For humans, knowing multiple languages (*multilinguality*) naturally implies knowing the correspondence between the words and phrases of the same meaning across those languages (*crosslinguality*). Indeed, when exposed to different linguistic environments, people can develop crosslingual capabilities by grounding the languages in physical world interactions. For example, we can relate the English world "apple" to the Spanish word "manzana" because in both linguistic environments the corresponding words refer to the same fruit in the real world. On the other hand, modern LLMs are trained purely based on the statistical relations in the text corpus without any grounding in the real world. In specific tasks such as machine translation, in order to teach the models to correlate notions across different languages, it is common to train with *parallel corpora* — collections of pairs of texts with the same meaning but in different languages (Eisenstein, 2019). However, as

the training process of widely used LLMs is often unknown, it is difficult to ascertain whether parallel corpora or other crosslingual supervision mechanisms were employed. This is particularly relevant for models that may naturally perform well in multiple languages due to their massive web data pretraining, even though they were not explicitly designed/advertised to target multilingual capabilities. [1] This ambiguity motivates our central research question: *How well do multilingual LLMs exhibit crosslingual capabilities?*

To state the problem more precisely, we define[2] the multilingual and crosslingual capabilities as follows. Denote an instance of a given task $\mathsf{T}$ as a tuple $(\mathcal{K}, \mathcal{C}, \mathcal{O})$, where $\mathcal{K}$ is the (optional) knowledge learned from training data, $\mathcal{C}$ is a context, and $\mathcal{O}$ is the correct answer. The *multilingual performance* on $\mathsf{T}$ measures the average performance across each language $\ell$ on an evaluation set $\{(\mathcal{K}_\ell, \mathcal{C}_\ell, \mathcal{O}_\ell)\}$ of task instances, where the subscript $\ell$ indicates the realization of the knowledge/context/answer in a specific language. On the other hand, the *crosslingual performance* on $\mathsf{T}$ measures the average performance on an evaluation set $\{(\mathcal{K}_\ell, \mathcal{C}_{\ell'}, \mathcal{O}_{\ell''})\}$ of crosslingual task instances, where $\ell, \ell', \ell''$ can be different languages.

For example, consider a task of REPEAT. The multilingual performance simply measures the model's capability of *copying* the context provided in different languages (i.e., $\mathcal{O}_\ell = \mathcal{C}_\ell$), whereas the crosslingual version of the task is equivalent to a much more challenging task of *translation* (i.e., $\mathcal{O}_{\ell''} = \text{Translate}_{\ell' \Rightarrow \ell''}(\mathcal{C}_{\ell'})$). Another example task is question-answering (QA), where the crosslingual version requires the model to apply *knowledge* $\mathcal{K}_\ell$ learned from one language $\ell$ to answer the question in a different language $\ell'$.

With those definitions, we summarize the main studies and contributions as below:

**Crosslingual capabilities** (§ 2): We formulate the question of multilingual vs crosslingual capabilities in LLMs. Through both translation tests (§ 2.1) and embedding distance evaluations (§ 2.2), we confirm that modern LLMs have strong crosslingual capabilities.

**Crosslingual knowledge barrier** (§ 3): We design crosslingual QA tasks, and observe a crosslingual knowledge barrier: LLMs have a significant performance gap on QA tasks formulated in a different language from the original language in which the knowledge is learned (see Fig. 1). Via extensive experiments across six models, we confirm a systematic presence of such barriers to knowledge learned both during the pretraining (§ 3.1) and fine-tuning (§ 3.2) stages.

**Towards overcoming the barrier** (§ 4): We propose a simple mixed-language training strategy (§ 4.2) and show that it can effectively reduce the knowledge barrier, outperform other baseline methods based on prompt engineering (§ 4.1), and further improve the few-shot learning performance. Furthermore, we show that even mixed-language training on out-of-domain data can be effective.

## 2 MULTILINGUAL LLMs HAVE COMPETITIVE CROSSLINGUAL CAPABILITIES

We demonstrate the crosslingual capabilities of existing multilingual LLMs from two perspectives: machine translation performance (§ 2.1) and an analysis of multilingual text embeddings (§ 2.2).

**Evaluation focus. (1) Languages**: We focus on five widely spoken languages: English (en), French (fr), German (de), Spanish (es), and Italian (it). Since our crosslingual study relies on the model being multilingual (i.e., that it already knows the languages well), we chose to evaluate these languages, as explicitly mentioned in the reports of some open-source models (Mistral, 2024). **(2) Multilingual LLMs:** We focus on six popular LLMs that have exhibited multilingual capabilities, including four open-source models: Llama2-7B, Llama2-13B (Touvron et al., 2023), Mistral-7B (Jiang et al., 2023), Llama3-8B (Meta, 2024), and two proprietary models: GPT-3.5 and GPT-4 (Achiam et al., 2023). § D.1 provides specifications for those LLMs.

While the above selected models and languages are our primary focus due to their popularity, we extend our evaluation to encompass **16 languages and 15 multilingual LLMs** in § 3 and 4, highlighting the broader implications of our main findings on the crosslingual knowledge barrier.

---

[1] Recent efforts mine massive parallel texts from the web (Schwenk et al., 2021), which may have been used in the pre-training datasets of some LLMs, particularly those designed with multilingual capabilities.

[2] We leave some of the terms mathematically vague, as long as they are not conceptually ambiguous. E.g., to measure the performance with a given correct answer, depending on the specific task format, we could either ask the model to generate the specific sequence of tokens or to rank the correct answer among multiple choices.

## 2.1 MACHINE TRANSLATION PERFORMANCE

**Setup.** To perform machine translation tasks with the open-source LLMs, we use the prompting format proposed by Xu et al. (2024). For proprietary LLMs, we use the prompting template suggested on their official webpages.[3] For reference we report two strong baselines: 1) NLLB-3.3B, the largest supervised encoder-decoder translation model from the NLLB family (Costa-jussà et al., 2022) trained on parallel corpus for 204 languages; and 2) Google Translate API. We report translation performance measured by the COMET score (Rei et al., 2020), a metric to predict human judgments of machine translation quality, on FLoRes-101 benchmark (Goyal et al., 2022) for two directions per language: en → X and X → en.

**Multilingual LLMs achieve competitive performance in machine translation.** As shown in Appendix Tb. 7, even though the evaluated multilingual LLMs are not directly trained on parallel corpora, their translation ability is quite competitive when compared to translation models explicitly trained on parallel corpora or industrial-grade translation APIs. For example, the gap is within 2.11 COMET score for X → en translation. Notably, these models generally perform better when translating X → en, but worse in the opposite direction, potentially suggesting that they are more proficient with the English translations. These results are consistent with previous papers that focus on improving machine translation with pretrained LLMs (Zhu et al., 2024; He et al., 2024; Xu et al., 2024). However, as we will show in later sections, our study focuses on the crosslingual transferability of knowledge learned in the model weights beyond the direct translation task.

## 2.2 EMBEDDING OF MIXED TRANSLATED SENTENCES

We further investigate the explicit crosslingual ability of multilingual LLMs by probing their text embeddings. Specifically, we aim to verify whether the embeddings for a given text in English are similar to the embeddings when some words are presented in different languages. The embedding is a single vector representing the average of the last layer's activations across all tokens in the sentence.

**Setup.** We randomly sample $1,000$ examples from the WikiText-103 corpus (Merity et al., 2017), creating two **versions** for each: (1) The original text in **English**; (2) **Mixed-language-translated**: for each word, with a probability of $p = 0.8$ it is unchanged; and with a probability of $1 - p$, the word is (independently) translated, using the Google Translate API, into a random language selected from the set {en, fr, de, es, it}. The choice of $p$ corresponds to a 0.2 probability that each word is replaced, aligning with the 5 languages we evaluate. That is, each word has a 0.16 probability of being translated into a non-English language. We then obtain sentence embeddings from the LLM for both versions of each example. To establish **baselines** for comparison, we consider two scenarios representing an "upper bound" on the distance when perturbations are *unrelated* to the original content: (1) **Random Token Replacement**: with a probability of $p = 0.16$, each token is replaced with a random different token from the vocabulary; and (2) **Random Token Dropout**: with a probability of $p = 0.16$, a token is completely masked out by disallowing any attention to it. $p = 0.16$ is chosen to align with token modification probability in mixed-language translation.

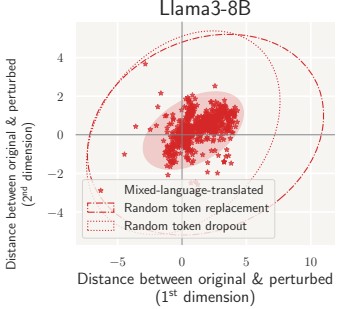

Figure 2: The embeddings of the English text and the mixed-language-translated text are closely aligned, unlike baselines with unrelated perturbations (e.g., random token replacement or dropout). The ellipses represent the covariance confidence intervals.

To visualize and compare embeddings, we reduce the original 4096-dimensional vectors to 2D using non-linear dimensionality reduction. We then calculate and visualize the per-coordinate distances between 2D embeddings of original English text and mixed-language translations, comparing these to baseline scenarios. Results for Llama3-8B are presented in Fig. 2, with other models' results deferred to Fig. 13.

**Embeddings of English and mixed-language-translated text are similar**, with difference vectors clustered near the origin. To quantify this, we conducted a two-sample statistical test comparing cosine similarities between: (1) original and mixed-translated sentence embeddings, and (2) original and random-token-replaced sentence embeddings. The resulting $p$-value ($< 0.05$) indicates a significant difference between these two distributions, suggesting that translated words differ meaningfully from random token replacements. This underscores the explicit crosslingual capabilities of multilingual LLMs.

---

[3]https://platform.openai.com/examples/default-translation

## 3  IDENTIFYING THE CROSSLINGUAL KNOWLEDGE BARRIER

While multilingual LLMs have demonstrated impressive *explicit* crosslingual abilities, such as performing translations for the input sequence given in the context, questions remain about their capacity to *implicitly* retrieve and utilize parametric knowledge stored in their weights across languages. For example, the model might be asked a question in one language (e.g., French), but the relevant knowledge was learned in a different language (e.g., English). As we will show in this section, LLMs struggle to seamlessly bridge the language gap when faced with tasks demanding implicit crosslingual knowledge transfer. We term this phenomenon the *crosslingual knowledge barrier*. In the following, we demonstrate the presence of such barriers for both general knowledge (§ 3.1) acquired during pretraining and domain-specific knowledge (§ 3.2) obtained through explicit fine-tuning.

### 3.1  CROSSLINGUAL KNOWLEDGE BARRIER IN GENERAL KNOWLEDGE

**Monolingual evaluation is inadequate for assessing crosslingual abilities.** Previous studies have evaluated open-domain Multiple Choice Question (MCQ) tasks on general knowledge in multilingual settings. For instance, the Mistral series models (Mistral, 2024) were benchmarked on translated versions of the Massive Multitask Language Understanding (MMLU) (Hendrycks et al., 2021) dataset in languages such as French (fr), German (de), Spanish (es), and Italian (it), separately. We refer to such monolingual evaluation setups as "full-translation". While such results indicate *multilingual* proficiency, they are insufficient to show *crosslingual* proficiency. The core issues is that the relevant general knowledge might be present in each of the evaluated languages in the pretraining dataset, so the LLMs could answer the full-translated questions based on knowledge learned in each individual language without invoking any crosslingual capabilities. Such possibility is difficult to verify, as the pretraining data for most LLMs are undisclosed.

**Mixed-language evaluation.** To directly invoke crosslingual capabilities of LLMs on general knowledge MCQ tasks, we suggest adopting an inherent crosslingual interaction approach through mixed-language MCQ formats. Specifically, we propose the following formats purposefully designed to be novel compositions unlikely to have been encountered during pretraining (examples in Fig. 3):

- Mixup translation: translating the question and all options into 5 *different* languages, with the language assignments randomly determined from the set {en, fr, de, es, it}.
- Question translation: translating the question into one non-English language.
- Options translation: translating all options into one non-English language.
- Question+GT-option translation: translating both the question and the ground truth option into one non-English language, while keeping the remaining options in English.
- GT-option translation: translating the ground truth option into one non-English language, while keeping the question and the rest of the options in English.
- One-wrong-option translation: randomly selecting one incorrect option and translating it into one non-English language.

In the above setups, even if a model has independently acquired knowledge in multiple languages, it will have to rely on crosslingual capabilities to select the correct answer. We perform translation via the Google Translate API, and all derived datasets have the same size as the original one.

As in the standard MCQ evaluation (Touvron et al., 2023; Zheng et al., 2024), we do the following: for open-source models, we calculate the likelihood of each option token and using the maximum one as the model prediction. For the closed-source models where token likelihoods are not accessible, we use the predicted best option (i.e., first token) with decoding temperature 0 as the answer. We additionally compare these two evaluation strategies on open-source models in Tb. 8 of the appendix.

**Crosslingual barrier in MMLU knowledge in 15 LLMs.** We focus on the MMLU benchmark for evaluating general knowledge, which comprises 4-option MCQs and includes $14k$ test samples from 4 domain categories (i.e., STEM, Social Sciences, Humanities, Others) across 57 subjects. The diversity of these domains enables us to draw general observations. In addition to the six models

Figure 3: Examples of original, full-translated, and proposed *mixed-language* multiple choice question (MCQ) formats.

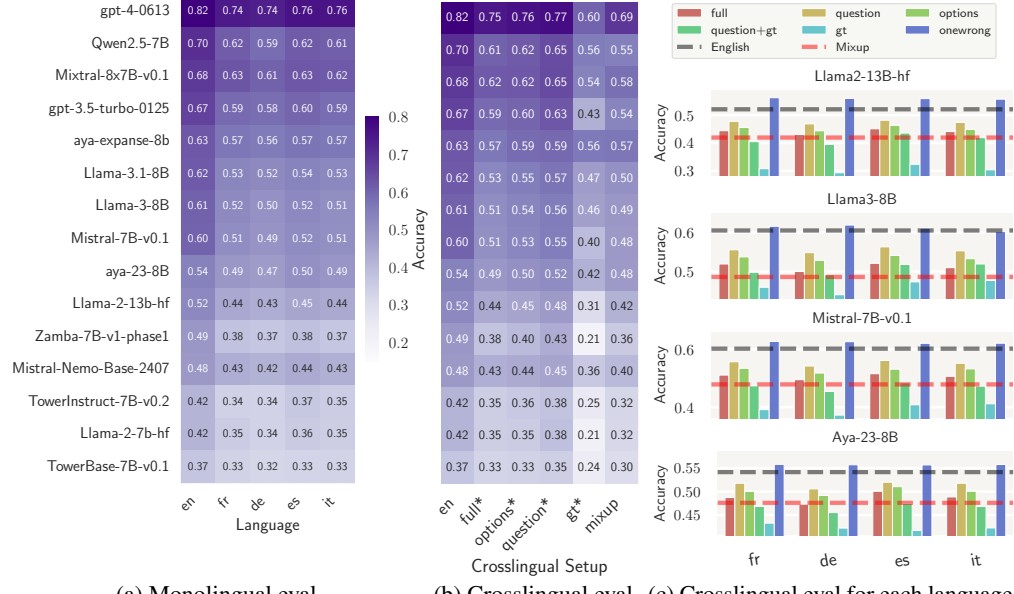

(a) Monolingual eval.  (b) Crosslingual eval.  (c) Crosslingual eval for each language.

Figure 4: **(a)** shows the monolingual evaluation on MMLU under 5 languages where 15 LLMs consistently perform better at answering multi-choice questions in English. Detailed results under four MMLU domains (STEM, Social Science, Humanities, Others) are in Fig. 14. **(b)** demonstrates the evaluation under cross-lingual settings, where * denote the average accuracy across {fr, de, es, it}. LLMs perform worse at answering MCQs in mixed-language settings than in English, especially the *ground truth option and mixup translation*, indicating the existence of cross-lingual knowledge barriers. **(c)** presents detailed cross-lingual evaluation results for each language on selected LLMs. We observe similar findings for all 15 LLMs in Fig. 15.

mentioned earlier, we evaluate 9 additional LLMs, including strong multilingual model Aya-23-8B, Aya-expanse-8b (Aryabumi et al., 2024), Llama-3.1-8B, Qwen2.5-7B, Mistral-Nemo, two Tower-series models trained under cross-lingual supervision, and two models beyond traditional Transformers—Zamba-7B, a state-space model (Glorioso et al., 2024), and Mistral-8x7B, a Mixture-of-Experts model (Jiang et al., 2024) **(1)** The traditional monolingual evaluation results in Fig. 4a show that all LLMs consistently achieve higher accuracy when MCQs are presented in English compared to other languages. This is likely because the relevant general knowledge is more frequently presented in English within the pretraining corpus, and LLMs struggle to transfer this knowledge to other languages automatically. **(2)** The results in Fig. 4b and Fig. 4c demonstrate a notable accuracy drop in the mixed-language settings, including question+GT-option, GT-option, and mixup translations, compared to monolingual settings (i.e., English and full-translation). This suggests that LLMs struggle to understand the more difficult contexts in multiple languages and to relate the corresponding parametric knowledge effectively to answer MCQs, highlighting a crosslingual knowledge barrier in the MMLU benchmark. We note such barrier exists even for the state-of-the-art models like GPT-4 (e.g., $81.82 \rightarrow 68.61$ when comparing English to mixup-translated MMLU). **(3)** The GT-option translation setting leads to the worst performance, indicating an inherent behavioral bias of LLMs that tends to avoid selecting a non-English option, even if it is the correct choice. This bias is further supported by the controlled comparisons in one-wrong-option translation settings, where LLMs achieve even higher accuracy than the English setting, as the model leverages the bias and avoids selecting the (incorrect) non-English option. **(4)** LLMs obtain higher accuracy on question-translated and options-translated settings than full-translated settings, likely because the MCQs under the former two settings still have remaining context in English, which helps the models perform better.

**Evaluation on 16 languages.** To demonstrate the universality of our findings, we extend our evaluation to 11 additional languages:

- Low-resource languages (Zhang et al., 2023c): Malay (ms), Danish (da), Finnish (fi), Norwegian (no), Bengali (bn), Amharic (am);
- Languages with token distributions significantly different from English: Russian (ru), Chinese (zn), Hebrew (he), Arbic (ar) and Hindi (hi).

The results in Fig. 5 show that the performance gaps between English (dashed line) and other languages persist in both monolingual (full translation) and mixed-language (option/gt-option translation) settings. This gap is particularly pronounced for low-resource languages such as Finnish (fi),

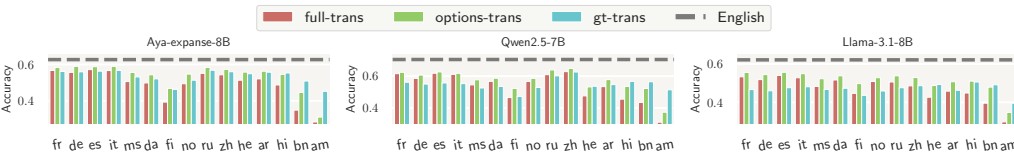

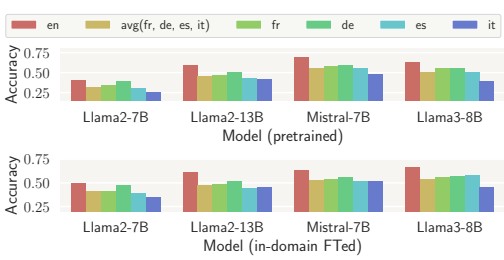

Figure 5: Evaluation across 16 languages reveals the universal crosslingual knowledge barriers in MMLU.

Figure 6: Multiple-choice accuracy of various multilingual LLMs on the Harry Potter Quiz benchmark before (top) and after (bottom) fine-tuning the model on in-domain content presented in English (i.e., Harry Potter-related documents selected from WikiText-103). Models consistently perform better at answering questions in English than in other languages, both before and after fine-tuning, indicating the presence of a crosslingual knowledge barrier.

Figure 7: Accuracy of pretrained LLMs on HP-Quiz across 16 languages. Models perform best in English, and perform better in high-resource languages (e.g., `fr`, `de`, `es`) than in low-resource ones (e.g., `bn`, `am`). Notably, Mistral-7B-v0.1 and Llama-3-8B show competitive performance compared to multilingual-focused models such as the Aya series and Tower series.

Bengali (`bn`), and Amharic (`am`), highlighting the universal challenge of cross-lingual knowledge barriers. Moreover, Llama-3.1-8B has a more balanced performance across various languages than Qwen2.5-7B and Aya-expanse-8B. For most non-English languages, multilingual models show the weakest performance when ground-truth options require cross-lingual reasoning.

## 3.2 CROSSLINGUAL KNOWLEDGE BARRIER IN DOMAIN-SPECIFIC KNOWLEDGE

In § 3.1, we demonstrated the crosslingual knowledge barrier for off-the-shelf LLMs in general knowledge required to solve MMLU tasks, where we assume this knowledge was obtained during pretraining. Here, we present a more controlled test through explicit fine-tuning on domain-specific knowledge. This experiment aims to answer the following question: *Could the model utilize the domain-specific knowledge (e.g., Harry Potter facts) acquired in one language (e.g., English) via fine-tuning to answer questions about this knowledge in other languages*? As we will show, the crosslingual knowledge barrier also exists for domain-specific knowledge.

**Harry Potter Quiz.** We use the Harry Potter world for the domain-specific knowledge evaluations, as it revolves around a highly detailed and extensive fictional universe with its own unique characters, terminology, and concepts. We manually curate a multiple-choice question-answering dataset called the Harry Potter Quiz (HP-Quiz) by extracting information from the Harry Potter Wiki pages[4]. Further details about the dataset are provided in § C.

**Evaluation.** Each multiple-choice question in the HP-Quiz dataset is available in five different languages: English (`en`), French (`fr`), German (`de`), Spanish (`es`), and Italian (`it`). To assess the crosslingual knowledge barrier, we consider both (1) the original model, and (2) the model fine-tuned on domain-specific corpora[5] presented only in English. For evaluation, we prompt the model with the multiple-choice question in each language, and report the accuracy of the model in selecting the correct answer in this multiple-choice task for each language.

**Crosslingual barrier also exists for Harry Potter knowledge.** As shown in Fig. 6, when presented with the same set of questions in 5 languages, the model consistently exhibits higher accuracy in answering correctly in English. This trend holds for both pretrained LLMs (left) and fine-tuned LLMs (right). After fine-tuning on domain-specific English corpora, despite the increase in model accuracy

---

[4]https://harrypotter.fandom.com/wiki/Main_Page

[5]Specifically, we preprocess the WikiText-103 dataset (Merity et al., 2017) and select documents highly relevant to the Harry Potter universe using a retriever (see § D.3 for details).

Table 1: Effect of inference-time mitigation methods evaluated on MMLU benchmarks. The highest accuracy achieved under the 0-shot/5-shot setting is underlined. ↓ denotes the accuracy drop observed in mixup MMLU compared to English MMLU. Simple prompt engineering cannot address the cross-lingual knowledge barrier problem. Although few-shot demonstrations enhance accuracy compared to the 0-shot setting, the performance gap between mixup MMLU and English MMLU remains significant. For reference, GPT-4 achieves 81.82 (0-shot) on English MMLU, and 68.61↓13.21 (0-shot), 73.58↓8.24 (5-shot english demonstrations), 77.71↓4.11 (5-shot biased demonstrations) on mixup MMLU.

| Eval setup | Prompt | Llama2-7B | Llama2-13B | Mistral-7B | Llama3-8B |
|---|---|---|---|---|---|
| English (0-shot) | A/B/C/D (default) | 41.53 | 52.11 | 60.21 | 60.54 |
| Mixup (0-shot) | A/B/C/D (default) | 32.18 ↓9.35 | 41.97 ↓10.14 | 47.86 ↓12.35 | 48.62 ↓11.92 |
| | a/b/c/d | 30.80 | 41.68 | 47.78 | 44.10 |
| | 1/2/3/4 | 27.96 | 38.39 | 45.56 | 44.63 |
| | Multilingual-Aware instruction 0 | 31.19 | 41.01 | 47.14 | 48.13 |
| | Multilingual-Aware instruction 1 | 31.23 | 41.35 | 46.80 | 47.89 |
| Mixup (5-shot) | English demonstrations | 35.23 | 43.15 | 49.46 | 50.99 |
| | Same bias demonstrations | 36.92 ↓4.61 | 44.32 ↓7.79 | 51.07 ↓9.14 | 51.65 ↓8.89 |
| | Translate-then-Answer demonstrations | 30.02 | 42.93 | 42.27 | 47.79 |

in English (which is more evident for Llama2-7B and Llama3-8B), the crosslingual knowledge barrier persists. This suggests that LLMs struggle to fully utilize the parametric knowledge acquired during English fine-tuning to answer related questions in other languages. These observations provide evidence that the crosslingual knowledge barrier extends beyond general knowledge into specific domains. To justify our studied models and languages, we present the results of 11 pretrained LLMs on HP Quiz across 16 languages in Fig. 7, where models perform best in English, and perform even worse low-resource ones (e.g., bn, am) than our studied high-resource languages (e.g., fr, de, es). Notably, Mistral-7B and Llama3-8B demonstrate competitive performance to some multilingual-focused models such as the Aya series and Tower series.

## 4 OVERCOMING CROSSLINGUAL KNOWLEDGE BARRIERS

In this section, we explore potential methods to overcome the manifest crosslingual knowledge barrier that we identified in the existing multilingual LLMs. We consider two types of potential mitigation methods, inference-time interventions (§ 4.1), including prompt engineering and few-shot demonstrations, and training-time interventions (§ 4.2), including mixed-language fine-tuning on general and domain-specific corpora.

### 4.1 INFERENCE-TIME MITIGATION

We evaluate inference-time mitigation methods to improve LLM performance on the mixup-translated MMLU, a challenging crosslingual setting evidenced by the low performance in Fig. 4b.

**Prompt engineering.** We evaluate the following prompting strategies: (1) **Alternative option ID characters**. We replace the default A/B/C/D with a/b/c/d or 1/2/3/4, motivated by recent evidence on selection bias in option IDs for MCQ tasks (Zheng et al., 2024) and to account for the possibility that the Arabic numerals are more invariant to languages. (2) **Multilingual awareness instruction**: We add an explicit instruction before the MCQs (e.g., "*Remember that the question and options can be in different languages*") to make models aware of the potential presence of other languages.

**Few-shot demonstrations.** Our evaluation mainly considers the 0-shot setting, which excludes any biases introduced by the few-shot demonstrations (Zhao et al., 2021), but we also conduct 5-shot experiments to further investigate crosslingual performance. MMLU covers 57 subjects, and the few-shot demonstrations for each subject are derived from the corresponding development set and shared across all test samples within the same subject. We employ several strategies to construct few-shot demonstrations: (1) **English** demonstration: English MCQ and answer pairs. (2) **Same bias** demonstration: mixup-translated MCQ and answer pairs, where each MCQ demonstration is constructed in the same way as the test sample. (3) **Translate-then-Answer** demonstration: For each mixup-translated MCQ, we prompt LLMs first to translate it into English before producing the final answer. To help LLMs follow the explicit translation instruction, we provide demonstrations where each includes a mixup-translated MCQ, the corresponding English MCQ, and its answer. We provide the detailed prompt templates in § D.2.

From the results in Tb. 1, (1) regarding prompt engineering, we observe no improvement and even a performance drop compared to the default prompt. It suggests that the crosslingual knowledge barrier

is an inherent failure of LLMs that cannot be effectively addressed by simple prompt engineering. (2) 5-shot settings consistently improve performance compared to 0-shot settings on mixup MMLU because providing demonstrations in the corresponding subject helps LLMs generalize to knowledge-intensive tasks. (3) Mixup demonstrations lead to better performance than English demonstrations because the mixed language pattern in the demonstrations matches that of the test examples. (4) Translate-then-Answer demonstrations are not effective. We observe failure patterns where, after translating to English, sometimes LLMs merely continue generating text without outputting the desired answer for the MCQ task. (5) Even under the best demonstration strategy, there still exists a substantial accuracy gap in mixup MMLU compared to English MMLU. Consequently, we explore training-time intervention in § 4.2 via mixed language fine-tuning methods.

### 4.2 MIXED-LANGUAGE FINE-TUNING

Given the limited success of inference-time interventions, we turn our attention to training-based methods that aim to directly instill better crosslingual knowledge in the model itself. Specifically, we explore mixed-language fine-tuning, where we explicitly construct a fine-tuning dataset comprising examples from multiple languages. To ensure a balanced representation of different languages, we split the training data into smaller units and randomly select a target language for each unit, translating the unit into that language if necessary. This approach also ensures that the translated data is of similar size as the original English data, enabling a fair comparison. Note that this approach differs from using parallel corpora as each unit is only presented in a single language.

We explore different choices for the smallest unit for translation, including the following settings:

- **Full document translation**: the entire document (example) is translated to a random language.
- **Sentence-level translation**: each document is split into units of sentences, using common English punctuation marks (Python regex `r'(\s*[\.,;!?]\s+)'`). Each sentence is then translated independently.
- **$k$-word chunk-level translation**: the document is split into chunks of $k$ words, where a "word" is any consecutive sequence of characters separated by one or more non-word characters defined by the Python regex `r'(\W+)'`. We found that the translation tool could be confused by $k$ words that span across sentence boundaries, so we did a little tweak by splitting into sentence first, and then split each sentence into $k$-word chunks.

Unless otherwise specified, for each translation unit, the target language is always randomly chosen uniformly from {`en`, `fr`, `de`, `es`, `it`}; for `en`, the translation is a No-Op.

We explore mixed-language fine-tuning of the original model on two types of corpora: general knowledge where we use WikiText-2 or WikiText-103 (Merity et al., 2017), and domain-specific where we use a subset of WikiText-103 that is highly related to Harry Potter based on BM25 similarity ranking (the details are deferred to § D.3).

**The differences between mixed-language fine-tuning and our cross-lingual evaluation setups** are noteworthy: i) In the general-knowledge evaluation, Mixup MMLU operates mixed-language multi-choice question at the option/question level, whereas mixed-language fine-tuning occurs at the document, sentence, or word levels. ii) In the domain-specific knowledge evaluation, we use the fully translated HP-Quiz (i.e., without mixup pattern) to evaluate the crosslingual capabilities[6], resulting in a more natural setting. iii) Additionally, we consider general Wikipedia documents as the fine-tuning corpus, which may not be directly related to MMLU/HP-Quiz tasks.

**Mixed-language fine-tuning on general corpus.** We first experiment with fine-tuning LLMs on a general corpus, WikiText-2 (keyword searching suggests that WikiText-2 has no overlap with Harry Potter characters or spells), with different choices of translation units. Specifically, we fine-tune the model for a single epoch, with a learning rate of $2 \times 10^{-5}$ and a batch size of 32, and report the multiple-choice accuracy on the Harry Potter Quiz. As shown in Fig. 8, the models fine-tuned on mixed translated general corpora achieve higher accuracy on HP-Quiz tasks than the model fine-tuned on the English corpus. This suggests that mixed-language fine-tuning could potentially help LLMs improve crosslingual capabilities: By exposure to frequent language switch during fine-tuning, LLMs can better adapt to the setting when the same knowledge is asked in a different (and usually non-English) language. Mixed-language FT also improves the performance on English HP-Quiz.

---

[6] In § 3.2, we have full control on the language in which the model learns the parametric knowledge.

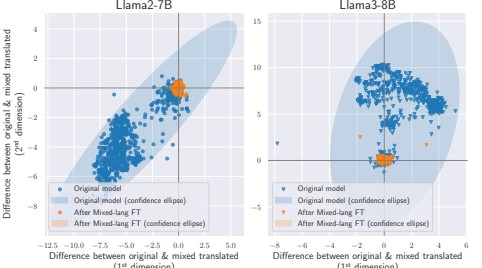

Figure 8: Fine-tuning on a mixed-language general corpus (e.g., WikiText-2) enhances the model's performance on domain-specific tasks (e.g., Harry Potter knowledge test) across multiple languages, including English. See Fig. 16 for results on Mistral-7B and Llama2-13B.

Table 2: Fine-tuning LLMs on the mixed-languages general corpus WikiText-103 can improve the performance on English and mixup MMLU benchmarks under 0-shot & 5-shot settings.

| Model | Llama2-7B | | Llama3-8B | |
| --- | --- | --- | --- | --- |
| | En MMLU | Mixup MMLU | En MMLU | Mixup MMLU |
| Un-FTed | 41.53 | 32.18 | 60.54 | 48.62 |
| En FTed | 41.21 | 31.46 | 60.32 | 47.83 |
| Mixed language (sentence) FTed | 42.05 | 34.08 | 60.45 | 51.75 |
| Mixed language (words) FTed | 42.00 | 34.06 | 60.28 | 50.88 |
| | (En demo) | (En demo \| Bias demo) | (En demo) | (En demo \| Bias demo) |
| Un-FTed + 5-shot | 45.88 | 35.23  36.92 | 65.00 | 50.99  51.65 |
| En FTed + 5-shot | 45.83 | 35.43  36.49 | 64.97 | 50.38  50.88 |
| Mixed language (sentence) FTed + 5-shot | 45.95 | 36.80  38.14 | 65.06 | 54.45  54.57 |
| Mixed language (words) FTed + 5-shot | 46.15 | 37.35  38.56 | 64.91 | 54.46  54.64 |

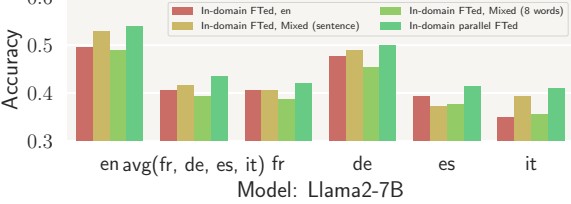

Figure 9: After mixed-lang FT (sentence), embeddings of original English text & mixed-language-translated text are more closely aligned, indicating a stronger knowledge correlation between En & other langs.

To further investigate how mixed-language FT improves crosslingual capabilities, we conduct a text embedding analysis with similar setups as in § 2. We examine, in the fine-tuned model, if the embeddings for a given English text are similar to the embeddings when some words are presented in different languages. The results show that the mixed-language (at sentence-level) fine-tuned model indeed has a much smaller text embedding distance compared to the original model, indicating that mixed-language FT can strengthen the knowledge correlation between English and other languages.

Our second experiment is fine-tuning LLMs on WikiText-103, a general corpus that offers a larger size and a broader range of knowledge compared to WikiText-2, and reporting the accuracy on MMLU variant benchmarks. (1) As shown in Tb. 2, fine-tuning on English WikiText-103 corpora hurts the performance, likely because it is an out-of-domain corpora for MMLU tasks. However, fine-tuning on mixed translated WikiText-103 corpora can lead to improvements, which are particularly noticeable on the mixup MMLU benchmarks. These results indicate that multiple language switches during fine-tuning enable LLMs to better understand and process multilingual inputs, become more robust to variations in language and phrasing, and perform better in knowledge-intensive crosslingual tasks. (2) Combining training-time interventions with test-time interventions can further enhance performance. While adding 5-shot biased demonstrations to our fine-tuned models leads to the best performance on mixup MMLU, adding 5-shot English demonstrations is also effective. This indicates the general applicability of our fine-tuned models across different scenarios. (3) Fine-tuning with both word-level and sentence-level mixed language WikiText-103 corpora effectively improves MMLU performance. Word-level mixing slightly outperforms in 5-shot settings, while sentence-level mixing is more effective in 0-shot settings. We defer the results on additional MMLU variant benchmarks to § E.3.

**Mixed-language fine-tuning on domain-specific corpus.** Similarly, we investigate the effectiveness of mixed-language fine-

Figure 10: Fine-tuning on a mixed-language domain-specific corpus (i.e., Harry Potter related documents from WikiText-103) generally enhances the performance on the Harry Potter Quiz dataset across multiple languages, including English.

tuning for the domain-specific task. Specifically, we fine-tune the model on mixed-language versions of in-domain corpora (i.e., Harry Potter-related documents from WikiText-103) and evaluate performance on the HP-Quiz. For an upper bound reference, we also report results from fine-tuning on a collection containing examples in all five languages (5× larger dataset size than our approach). As shown in Fig. 10, mixed-language fine-tuning (especially at sentence-level) can lead to better overall performance on HP-Quiz compared to English fine-tuning.

**Mixed-language fine-tuning helps the QA performance on out-of-distribution languages.** We evaluated our fine-tuned models on languages that were not included in fine-tuning data. Results in Fig. 11 show that mixed-language (with {en, fr, de, es, it}) fine-tuning on general Wiki corpus can improve the cross-lingual performance of 11 other languages on HP-Quiz, including low-resource ones and those substantially different from English. Furthermore, as shown in Fig. 12, mixed-language fine-tuning also boosts the performance of MMLU variants in various cross-lingual settings for four low-resource languages.

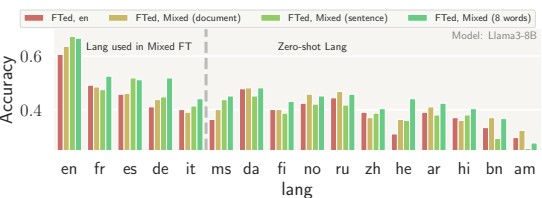 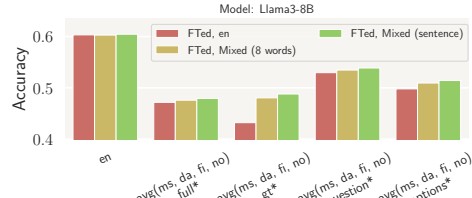

Figure 11: Mixed-language FT on WikiText-2 with {en, fr, de, es, it} enhances accuracy on Harry Potter Quiz across other languages that are not used during fine-tuning. Such improvements incur in low resource languages (e.g., ms, bn, am) and languages that are rather different from English (e.g., zh, ru, he, ar, hi) with low amount of shared tokens.

Figure 12: Mixed-language FT on Wiki-103 with {en, fr, de, es, it} enhances accuracy on MMLU variants across low-resource languages {ms, da, fi, no} that are not used during fine-tuning.

## 5 RELATED WORK

Understanding language models' performance in multilingual settings is an active area of research. Prior works have identified strong variations in the amount of knowledge across different languages, attributed to differences in training corpora sizes (Jiang et al., 2020; Kassner et al., 2021; Ryan et al., 2024). These insights have been used to improve model performance, such as leveraging multilingual self-consistency (Ohmer et al., 2023). Efforts have also been devoted to studying well-established tasks for monolingual models in crosslingual scenarios, such as crosslingual pretraining (Lample & Conneau, 2019; Abadji et al., 2022; Schioppa et al., 2023), information retrieval (Yu et al., 2021), knowledge editing (Wang et al., 2024a; Xu et al., 2023a; Beniwal et al., 2024; Xu et al., 2023b), text summarization (Wang et al., 2023a; Huang et al., 2023) and instruction tuning and alignment (Chirkova & Nikoulina, 2024b; Zhang et al., 2023b; Ranaldi et al., 2023; Zhu et al., 2023; Wu et al., 2024). The work most related to ours is Qi et al. (2023), which proposed a metric to evaluate multilingual models' factual knowledge consistency across languages. One key difference is that their study does not account for different factors that could contribute to the crosslingual consistency; while we formulate a controlled setting of crosslingual knowledge barrier, measuring precisely the ability to transfer knowledge learned (only) in one language to another language. Furthermore, we proposed mitigation methods to reduce such knowledge barrier.

We refer the readers to § B for a more comprehensive discussion of related work.

## 6 CONCLUSION AND FUTURE WORK

In this work, we observed that despite the competitive performance of multilingual LLMs in explicit crosslingual tasks such as translation, those models fail to transfer learned knowledge across the language boundary, a phenomenon we termed as the crosslingual knowledge barrier. Through comprehensive evaluations on both general and domain-specific knowledge, we confirmed a systematic presence of such barriers across all 6 models and the five languages that those models know. Finally, we evaluated both test-time and training-time mitigations and proposed a simple and effective mixed-language fine-tuning procedure to reduce the knowledge barrier in those models. We discuss our limitations and further work in § A.

## REPRODUCIBILITY STATEMENT

In this paper, we have taken steps to ensure the reproducibility of our results: (1) The source code for our evaluation and fine-tuning is available in the supplementary material. We have provided the README and scripts to replicate the experiments in the paper. (2) The details about the Harry Potter Quiz dataset, and the WikiText-103 subset on Harry Potter related documents are provided in § C. We used the public MMLU, WikiText-2, WikiText-103 datasets. The Mixup-version MMLU and English-version HP-Quiz datasets are provided in the supplementary material. (3) We described the experimental setups and hyper-parameters in each section (§ 2 to 4). More details on models, LLM evaluation, fine-tuning and computation resources are included in the Appendix § D.

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

APPENDIX

## A  LIMITATIONS AND FUTURE WORK

One limitation of our work is that all translations were performed using Google Translate instead of human experts. While Google Translate is recognized as a high-quality industrial translation service, validating and enhancing the translation quality remains an area for future work.

One important question that is not answered in this paper is how these models develop the crosslingual capabilities (despite the existence of crosslingual knowledge barriers). This is intriguing because unlike humans exposed to multiple lingual environments, LLMs do not have grounding in the physical world to help them establish connections between different words that refer to the same thing in the real world. While there is some preliminary work on grounding LLMs with the physical world (e.g., Zhang et al., 2023a; Wang et al., 2023b; Gao et al., 2024; Cheng et al., 2024), the majority of the LLMs nowadays are still trained via next-word prediction without interaction with the physical world. Therefore, an interesting future direction is to understand the mechanisms that allow the LLMs to develop crosslingual capabilities.

Another interesting observation is that mixed-language fine-tuning (on out-of-domain data) can improve the question-answering performance on both non-English languages and English in many of our evaluations. Previous studies (e.g., Abutalebi et al., 2008; Bialystok et al., 2012; Marian & Shook, 2012) have shown that multilinguality could have a positive effect on human cognitive abilities. But how does better crosslingual capabilities impact LLMs' reasoning abilities (in English) remains to be fully understood.

## B  EXTENDED RELATED WORK

**Understanding and improving multilingual LMs.**  Understanding language models' performance in multilingual settings is an active area of research. Prior works have identified strong variations in the amount of knowledge across different languages, attributed to differences in training corpora sizes (Jiang et al., 2020; Kassner et al., 2021; Ryan et al., 2024). These observations have also been leveraged to improve models' performance, especially in English. For instance, Ohmer et al. (2023) propose using multilingual self-consistency of predictions to assess how well the model understands a given task. Wu et al. (2024) suggest using a reward model in a different language during fine-tuning for alignment from human feedback can yield better-aligned models than using one in the same language as the pre-trained model. Efforts have also been devoted to studying well-established tasks for monolingual models in crosslingual scenarios, such as crosslingual pretraining (Lample &

Conneau, 2019; Abadji et al., 2022; Schioppa et al., 2023), information retrieval (Yu et al., 2021), knowledge editing (Wang et al., 2024a; Xu et al., 2023a; Beniwal et al., 2024; Xu et al., 2023b), text summarization (Wang et al., 2023a; Huang et al., 2023) and instruction tuning (Chirkova & Nikoulina, 2024b; Zhang et al., 2023b; Ranaldi et al., 2023; Zhu et al., 2023).

The closest work to ours is Qi et al. (2023), which proposes a metric to evaluate the consistency of a multilingual language model's factual knowledge across languages. They find that while increasing model size generally leads to higher factual accuracy in most languages, it does not necessarily improve crosslingual knowledge consistency. One key difference is that their study does not account for different factors that could contribute to the crosslingual consistency (e.g., a model independently learns the knowledge in both languages during pretraining could lead to a high consistency); while we formulate a controlled setting of crosslingual knowledge barrier, measuring precisely the ability to transfer knowledge learned (only) in one language to another language. Furthermore, we also proposed mitigation methods that could effectively reduce the knowledge barrier.

**Machine translation ability of LLMs.** The off-the-shelf pretrained LLMs show promise in machine translation but still lag behind the commercial translation system, especially in low-resource languages. Previous studies have sought to enhance LLM translation capabilities through various prompting and fine-tuning methods. Zhu et al. (2024) introduce crosslingual translation in-context examples, while He et al. (2024) employ advanced prompt engineering that induces translation-related knowledge (e.g., keywords, topics) from the given source sentence to guide the final translation process. Xu et al. (2024) propose a two-stage fine-tuning approach, first enhancing proficiency in non-English languages by fine-tuning on non-English monolingual data, and then fine-tuning on high-quality parallel data for translation task. Our work has a different goal of comprehensively examining LLMs' crosslingual capabilities, beyond the translation task. We show that even though LLMs are very competitive at explicit translation tasks, they could struggle in more demanding tasks that requires implicit knowledge transfer across language boundaries.

**Crosslingual transfer of multilingual models.** Crosslingual transfer refers to transfer learning that fine-tunes the model on a target task in one language (e.g., English), and then makes predictions for this task in another, typically more low-resource language. It addresses the challenges of limited training data in the target language for a target task. It has been broadly studied for natural language understanding (Schioppa et al., 2023; Artetxe et al., 2020; Pires et al., 2019; Wu & Dredze, 2019; Li et al., 2022) and generation tasks (Chirkova & Nikoulina, 2024a; Bapna & Firat, 2019; Vu et al., 2022; Maurya et al., 2021; Li & Murray, 2023; Tanwar et al., 2023) for multilingual models such as mBART, mT5, NLLB family. For instance, Chirkova & Nikoulina (2024a) demonstrated that fine-tuning the full model with a small learning rate yields the best crosslingual language generation performance, outperforming other methods such as adapter (Bapna & Firat, 2019), prompt-tuning (Vu et al., 2022)) and hyperparameter tuning (Chirkova & Nikoulina, 2024a). Additionally, several studies have improved crosslingual generalization by mixing auxiliary unsupervised data from additional languages during fine-tuning. For example, sampling target language examples with probability (e.g., 1%) when forming the mini-batch (Chirkova & Nikoulina, 2024a; Vu et al., 2022).

Our study focuses on more recent autoregressive LLMs (e.g., Llama series, Mistral, GPT-3.5, GPT-4) that acquire multilingual capabilities from their internet-scale pretraining corpora. While several works have explored approaches to enhancing LLMs' crosslingual transfer abilities such as fine-tuning with adapter merging (Zhao et al., 2024), our work differs in its primary focus. We aim to provide a comprehensive understanding of the crosslingual capabilities of pretrained LLMs on tasks requiring explicit (e.g., translation tasks) and implicit crosslingual transfer (e.g., question-answering tasks involving general or domain-specific knowledge). Furthermore, to improve crosslingual transfer ability of LLMs in general, our study employs fine-tuning on (out-of-domain) general corpora and proposes a principled approach to processing mixed language data at different levels of granularity, including word, sentence, and document levels.

**Compositional generalization.** We also acknowledge that the crosslingual knowledge barrier can be viewed as an instance of the broader challenge of compositional generalization (Lake et al., 2017; Kim & Linzen, 2020; Hupkes et al., 2020; Xu et al., 2022; Yu et al., 2024) — the ability to systematically combine different skills to understand and produce novel compositions not directly trained on. In the case of crosslingual knowledge understanding, models must compose the skills of

question answering and knowledge translation. However, this specific combination of crosslingual knowledge consistency warrants dedicated study due to its strong practical implications, as ensuring consistent feedback across languages is crucial for deploying trustworthy and effective multilingual AI assistants to a global user base.

**Behavioral bias of LLMs.** Recent research also studies various behaviors and biases in LLMs that are different from human reasoning, such as reversal curse (Berglund et al., 2024; Grosse et al., 2023), order and position bias (Wang et al., 2024b; Pezeshkpour & Hruschka, 2024), option ID bias in multiple-choice question tasks (Zheng et al., 2024), premise order bias (Chen et al., 2024), susceptibility to distraction by irrelevant context (Shi et al., 2023). These studies provide a deeper understanding of LLMs and suggest various ways to improve those models. Our paper contributes to this important line of research from the perspective of crosslingual behaviors.

**Code-switching.** Code-switching training (Yang et al., 2020; Song et al., 2019) uses parallel text to teach models the relation between original and translated tokens, primarily for machine translation. Compared to code-switching, our proposed mixed-language fine-tuning does not create parallel text, and aims to encourage LLMs to cross language barriers without requiring architectural changes or special handling of parallel text. Our approach can directly handle multiple languages while maintaining a similar number of tokens as the original dataset.

## C  THE HARRY POTTER QUIZ DATASET

We use Harry Potter as a setting to mimic domain-specific knowledge, as it revolves around a highly detailed and extensive fictional universe with its own unique characters, terminology, and concepts. We manually curate an English-only dataset named Harry Potter Quiz (or HP-Quiz in short) by collecting information about characters and magic spells[7] from the Harry Potter Wiki pages[8]. For characters, we gather attributes such as gender, hair color, house[9], and relationships with other characters. Regarding magic spells, we collected data on the types of spells they belong to. We then curate multiple-choice questions and answers based on the collected information. Specifically, the dataset consists of 300 questions in total, 157 questions about characters and 143 questions about magic spells. We format these questions as multiple choice questions.

Below is the full list of characters and spells included in HP-Quiz:

**25 Characters**  Aberforth Dumbledore, Albus Potter, Ariana Dumbledore, Arthur Weasley, Astoria Malfoy, Cedric Diggory, Charles Weasley, Cho Chang, Draco Malfoy, Dudley Dursley, Euphemia Potter, Fleamont Potter, Harry Potter, Hermione Granger, James Potter I, Kendra Dumbledore, Lily J. Potter, Lucius Malfoy, Narcissa Malfoy, Percival Dumbledore, Petunia Dursley, Roger Davies, Ron Weasley, Scorpius Malfoy, William Weasley

**143 Spells**  Aberto, Accio, Age Line, Alarte Ascendare, Alohomora, Anti-Cheating Spell, Anti-Apparition Charm, Anti-Disapparition Jinx, Anti-intruder jinx, Aparecium, Appare Vestigium, Apparition, Aqua Eructo, Arania Exumai, Arresto Momentum, Arrow-shooting spell, Ascendio, Avada Kedavra, Avifors, Avenseguim, Babbling Curse, Badgering, Bat-Bogey Hex, Bedazzling Hex, Bewitched Snowballs, Bluebell Flames, Blue sparks, Bombarda, Bombarda Maxima, Bravery Charm, Bridge-conjuring spell, Broom jinx, Bubble-Head Charm, Bubble Spell, Calvorio, Cantis, Capacious extremis, Carpe Retractum, Cascading Jinx, Caterwauling Charm, Cave inimicum, Celescere, Cheering Charm, Circumrota, Cistem Aperio, Colloportus, Colloshoo, Colovaria, Confringo, Confundo, Conjunctivitis Curse, Cracker Jinx, Cribbing Spell, Crinus Muto, Crucio, Defodio, Deletrius, Densaugeo, Deprimo, Depulso, Descendo, Deterioration Hex, Diffindo, Diminuendo, Dissendium, Disillusionment Charm, Draconifors, Drought Charm, Duro, Ear-shrivelling Curse, Ebublio, Engorgio, Entrail-Expelling Curse, Epoximise, Erecto, Evanesce, Evanesco, Everte Statum, Expecto Patronum, Expelliarmus, Expulso, Extinguishing Spell, Feather-light charm, Fianto Duri, Fidelius Charm, Fiendfyre, Finestra, Finite Incantatem, Finger-removing jinx, Firestorm, Flagrante Curse, Flagrate, Flame-Freezing Charm, Flask-conjuring spell, Flintifors, Flipendo, Flipendo Duo, Flipendo Maxima, Flipendo Tria, Flying charm, Fracto Strata, Fumos, Fumos Duo, Furnunculus, Fur spell, Geminio, Glacius, Glacius Duo, Glacius Tria, Glisseo, Gripping Charm, Hair-thickening Charm, Herbifors, Herbivicus, Homenum Revelio, Homonculous Charm, Hurling Hex, Impedimenta, Imperio, Inanimatus Conjurus, Incarcerous, Inflatus, Jelly-Brain Jinx, Jelly-Fingers Curse, Knee-reversal hex, Langlock, Lapifors, Leek Jinx, Levicorpus, Liberacorpus, Locomotor Mortis, Melofors, Meteolojinx recanto, Mimblewimble, Multicorfors, Obscuro, Oppugno, Orbis, Orchideous, Pepper Breath, Petrificus Totalus, Piscifors, Point Me

---

[7]In Harry Potter, the magic spell is a magical action used by witches and wizards to perform magic.

[8]https://harrypotter.fandom.com/wiki/Main_Page

[9]Hogwarts, the fictional boarding school of magic in the Harry Potter book series, is divided into four houses: Gryffindor, Slytherin, Ravenclaw, and Hufflepuff.

## D EXPERIMENTAL DETAILS

### D.1 EVALUATED MODELS

Tb. 3 provides the details of the models evaluated in our study.

Table 3: HuggingFace links or endpoint specifications for evaluated models.

| Model | Link |
|---|---|
| Llama2-7B | https://huggingface.co/meta-llama/Llama-2-7b-hf |
| Llama2-13B | https://huggingface.co/meta-llama/Llama-2-13b-hf |
| Mistral-7B | https://huggingface.co/mistralai/Mistral-7B-v0.1 |
| Llama3-8B | https://huggingface.co/meta-llama/Meta-Llama-3-8B |
| GPT-3.5 | https://platform.openai.com/docs/models/gpt-3-5-turbo, gpt-3.5-turbo-0125 endpoint |
| GPT-4 | https://platform.openai.com/docs/models/gpt-4-turbo-and-gpt-4, gpt-4-0613 endpoint |
| Aya-23-8B | https://huggingface.co/CohereForAI/aya-23-8B |
| Zamba-7B | https://huggingface.co/Zyphra/Zamba-7B-v1-phase1 |
| Mistral-8x7B | https://huggingface.co/mistralai/Mixtral-8x7B-v0.1 |

Table 4: Prompt templates for inference-time mitigation methods in mixup-translated MMLU evaluation. The templates are consistent across different evaluation setups, varying only in the language pattern of multiple-choice questions.

| Setting | Type | Prompt |
|---|---|---|
| 0-shot | Default prompt | The following are multiple choice questions (with answers) about {subject}.
{Mixup_MultiChoiceQuestion}
Answer: |
| | Multilingual-Aware instruction 0 | The following are multiple choice questions (with answers) about {subject}. Keep in mind that the question and options may be presented in various languages.
{Mixup_MultiChoiceQuestion}
Answer: |
| | Multilingual-Aware instruction 1 | The following are multiple choice questions (with answers) about {subject}. Remember that the question and options can be in different languages.
{Mixup_MultiChoiceQuestion}
Answer: |
| few-shot | English demonstrations | The following are multiple choice questions (with answers) about {subject}.
{En_MultiChoiceQuestion_Demo1}
Answer: {Answer_Demo1}
....
{Mixup_MultiChoiceQuestion}
Answer: |
| | Same bias demonstrations | The following are multiple choice questions (with answers) about {subject}.
{Mixup_MultiChoiceQuestion_Demo1}
Answer: {Answer_Demo1}
....
{Mixup_MultiChoiceQuestion}
Answer: |
| | Translate-Then-Answer demonstrations | The following are multiple choice questions (with answers) about {subject}. Remember that the question and options can be in different languages. First translate them all to English. Then output the answer.
Question: {Mixup_MultiChoiceQuestion_Demo1}
Answer:
Translate the question and options into English, and then answer.
Question: {En_MultiChoiceQuestion_Demo1}
Answer: {Answer_Demo1}
....
Question: {Mixup_MultiChoiceQuestion}
Translate the question and options into English, and then answer.
Question: |

### D.2 EVALUATION AND TRAINING DETAILS

**LLM Evaluation** For MMLU evaluation, we follow the templates in its official codebase[10] to construct the prompts for 0-shot and 5-shot settings. We employ a temperature of 0 for GPT-3.5 and

---

[10] https://github.com/hendrycks/test

GPT4, and we select the choice with the highest logits score as the predicted answer for open-source models. We provide the prompt templates for inference-time mitigation methods in Tb. 4.

For the Harry Potter evaluation, we use the following prompt template, with an example shown below:

```
 The following are multiple choice questions (with answers) about Harry Potter.

 Which house is Harry Potter belong to?
 A.Ravenclaw
 B.Slytherin
 C.Gryffindor
 D.Hufflepuff
```

After querying the model, we select the choice with the highest logical score as the predicted answer.

**LLM Fine-tuning** (1) For WikiText-103 fine-tuning, we fine-tune Llama3-8B for 200 steps, and Llama2-7B for 400 steps, with a learning rate of $2 \times 10^{-5}$ and a batch size of 32. (2) For fine-tuning on WikiText-2 or Harry Potter related documents from WikiText-103, we fine-tune the models for one epoch with the same set of hyperparameters. We use AdamW (Loshchilov & Hutter, 2018) as the optimizer.

**Computation Resources** All fine-tuning experiments are conducted on 2 NVIDIA A100 GPU cards, each with 80GB of memory. For the fine-tuning experiments, each training step takes 5.2 seconds for the Llama2-7B model and 6.1 seconds for the Llama3-8B model, with a batch size of 32. All LLM evaluation experiments can be conducted on one NVIDIA RTX A6000 GPU card with 48 GB of memory.

### D.3 WIKITEXT-103 SUBSET: HARRY POTTER-RELATED DOCUMENTS

We employ the BM25 algorithm (Trotman et al., 2014) (BM stands for best matching) for document ranking[11], which is a bag-of-words retrieval function that ranks documents based on the presence of query terms in each document. The WikiText-103 corpus comprises $M = 1,165,029$ documents $d_i, i \in [M]$. We concatenate the passages crawled from Harry Potter Wiki pages into a single document to use as a query $q$. We then calculate the similarity score between the query and each document in WikiText-103, denoted as $s_i = \text{Sim}(d_i, q)$. The top $k = 3$ relevant documents are listed in Tb. 5.

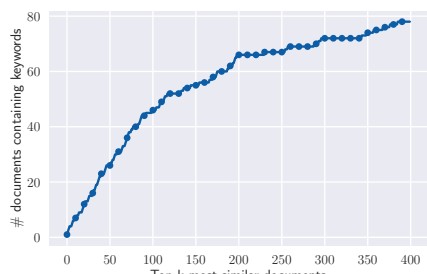

Table 6: The top $k$ retrieved documents containing the Harry potter keywords.

Additionally, we use the list of Harry Potter character names and spell names[12] as keywords to evaluate the quality of the retrieved documents and to identify additional relevant documents. Tb. 6 illustrates the trend that as $k$ increases, more documents containing the keywords are retrieved. Note that keyword matching is not a golden retrieval method and it only serves as reference because: (1) documents may not contain the full name of characters or spells (e.g., "Harry" instead of "Harry Potter"); (2) some spell names are generic and have multiple meanings (e.g., "Pack", "Avis").

Therefore, we combine the top documents retrieved by BM25 with keyword matching to create our final dataset. The final dataset contains 4,348 documents (0.37% of WikiText-103), comprising: (1) the top $k = 2000$ documents retrieved by BM25. Of these, 106 documents contain at least one exact keyword. (2) An additional 2,358 documents that contain the keywords.

---

[11]https://pypi.org/project/rank-bm25/

[12]The spell name "None" is excluded due to its generic nature.

Table 5: Top three most relevant documents to the Harry Potter universe in WikiText-103 based on BM25 document ranking. Keywords related to Harry Potter universe are **bolded**.

| 1 | In Philosopher's Stone, Harry re @-@ enters the wizarding world at age 11 and enrols in Hogwarts School of Witchcraft and Wizardry. He makes friends with fellow students **Ron Weasley** and **Hermione Granger**, and is mentored by the school's headmaster, Albus Dumbledore. He also meets Professor Severus Snape, who intensely dislikes and bullies him. Harry fights Voldemort several times while at school, as the wizard tries to regain a physical form. In Goblet of Fire, Harry is mysteriously entered in a dangerous magical competition called the Triwizard Tournament, which he discovers is a trap designed to allow the return of Lord Voldemort to full strength. During Order of the Phoenix, Harry and several of his friends face off against Voldemort's Death Eaters, a group of Dark witches and wizards, and narrowly defeat them. In Half @-@ Blood Prince, Harry learns that Voldemort has divided his soul into several parts, creating " horcruxes " from various unknown objects to contain them; in this way he has ensured his immortality as long as at least one of the horcruxes still exists. Two of these had already been destroyed, one a diary destroyed by Harry in the events of Chamber of Secrets and one a ring destroyed by Dumbledore shortly before the events of Half @-@ Blood Prince. Dumbledore takes Harry along in the attempt to destroy a third horcrux contained in a locket. However the horcrux has been taken by an unknown wizard, and upon their return Dumbledore is ambushed and disarmed by **Draco Malfoy** who cannot bring himself to kill him, then killed by Snape. |
|---|---|
| 2 | Luna, Ron, Ginny, and Neville join them in the forest and all six fly to the Ministry on , expecting to find and rescue Sirius. Once in the Department of Mysteries, Harry realises that his vision was falsely planted by Voldemort; however, he finds a glass sphere that bears his and the Dark Lord's names. Death Eaters led by **Lucius Malfoy** attack in order to capture the sphere, which is a recording of a prophecy concerning Harry and Lord Voldemort, which is revealed to be the object Voldemort has been trying to obtain for the whole year, the Dark Lord believing that there was something he missed when he first heard the prophecy. Lucius explains that only the subjects of the prophecies, in this case Harry or Voldemort, can safely remove them from the shelves. Harry and his friends, soon joined by members of the Order, enter a battle with the Death Eaters. Amidst the chaos, Bellatrix Lestrange kills Sirius and Harry faces Voldemort. Voldemort attempts to kill Harry, but Dumbledore prevents him and fights the Dark Lord to a stalemate. In the midst of the duel, Voldemort unsuccessfully tries to possess Harry in an attempt to get Dumbledore to kill the boy. Dumbledore does not do so and Voldemort escapes just as Cornelius Fudge appears, finally faced with first @-@ hand evidence that Voldemort has truly returned. |
| 3 | During another summer with his Aunt Petunia and Uncle Vernon, **Harry Potter** and Dudley are attacked. After using magic to save Dudley and himself, Harry is expelled from Hogwarts, but the decision is later rescinded. Harry is whisked off by a group of wizards to Number 12, Grimmauld Place, the home of his godfather, Sirius Black. The house also serves as the headquarters of the Order of the Phoenix, of which Mr. and Mrs. Weasley, Remus Lupin, Mad @-@ Eye Moody, and Sirius are members. **Ron Weasley** and **Hermione Granger** explain that the Order of the Phoenix is a secret organisation led by Hogwarts headmaster Albus Dumbledore, dedicated to fighting Lord Voldemort and his followers, the Death Eaters. From the members of the Order, Harry and the others learn that Voldemort is seeking an object that he did not have prior to his first defeat, and assume this object to be a weapon of some sort. Harry learns that the Ministry of Magic, led by Cornelius Fudge, is refusing to acknowledge Voldemort's return because of the trouble that doing so would cause, and has been running a smear campaign against him and Dumbledore. |

# E    ADDITIONAL EXPERIMENTAL RESULTS

## E.1    CROSSLINGUAL CAPABILITIES OF LLMS

**Machine translation**    Tb. 7 report the COMET score on FLoRes-101 benchmark (Goyal et al., 2022) for two directions per language: en → X and X → en. It shows that multilingual LLMs;s translation ability is quite competitive when compared to translation models explicitly trained on parallel corpora or industrial-grade translation APIs.

**Embedding analysis**    As shown in Fig. 13, for the four LLMs multilingual, including Llama2-7B, Llama2-13B, Mistral-7B and Llama3-8B[13], the embeddings of the original text and its mixed-translated counterpart exhibit a high degree of similarity, with their difference vectors clustering around the origin. This observation stands in stark contrast to the scenario where English words are replaced with random tokens. It implies the explicit crosslingual capabilities of the multilingual LLMs.

---

[13]We focus primarily on open-source models due to the cost associated with querying embeddings from proprietary models.

Table 7: COMET scores for machine translation tasks evaluated on FloRes-101 benchmark using multilingual LLMs (Llama2-7B, Mistral-7B, Llama2-13B, Llama3-8B, GPT-3.5, and GPT-4), models trained on parallel corpora (NLLB-3.3B), and an industrial-grade translation API (Google Translate). Multilingual LLMs achieve competitive translation performance against dedicated translation models and the translation API.

| | English (en) → other languages | | | | | Other languages → English (en) | | | | |
|---|---|---|---|---|---|---|---|---|---|---|
| | en → de | en → fr | en → es | en → it | Avg | de → en | fr → en | es → en | it → en | Avg |
| **Llama2-7B** | 81.67 | 84.54 | 84.76 | 85.17 | 84.04 | 87.61 | 87.96 | 85.60 | 86.47 | 86.91 |
| **Llama2-13B** | 71.63 | 79.91 | 81.00 | 76.68 | 77.81 | 88.26 | 88.91 | 85.83 | 86.99 | 87.50 |
| **Llama3-8B** | 73.89 | 81.19 | 81.03 | 81.16 | 79.32 | 88.52 | 88.61 | 86.45 | 87.03 | 87.65 |
| **Mistral-7B** | 76.18 | 78.46 | 80.04 | 76.64 | 77.83 | 87.73 | 88.05 | 85.72 | 86.11 | 86.90 |
| **GPT-3.5** | 87.53 | 86.97 | 86.40 | 86.46 | 86.84 | 89.14 | 89.42 | 87.47 | 88.41 | 88.61 |
| **GPT4** | 88.18 | 88.13 | 86.64 | 85.33 | 87.07 | 89.71 | 89.56 | 87.41 | 88.14 | 88.71 |
| **NLLB-3.3B** | 87.33 | 87.44 | 86.88 | 88.26 | 87.48 | 79.65 | 87.44 | 85.64 | 84.13 | 84.72 |
| **Google Translate** | 89.39 | 89.22 | 87.23 | 89.37 | 88.80 | 90.01 | 89.92 | 87.55 | 88.54 | 89.01 |

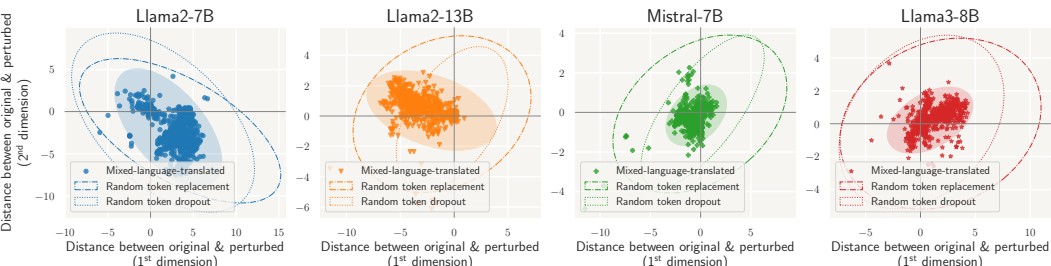

Figure 13: The embeddings of the original English text and the mixed-language-translated text are closely aligned, unlike baselines with unrelated perturbations (e.g., random token replacement or dropout). The ellipses represent the covariance confidence intervals.

### E.2 CROSSLINGUAL KNOWLEDGE BARRIERS IN LLMs

**Evaluation strategy**   We follow prior work on LLM evaluation (Zheng et al., 2024) to use two evaluation strategies: **(1)** Open-source: access the output probabilities of option ID tokens A/B/C/D and predict argmax. **(2)** Closed-source: compare the golden answer with the 1st generated token (decoding temperature=0), as the logits are not available for most closed-source models.

we evaluated the open-source LLMs using the 1st token produced as the answer in Tb. 8. The two evaluation strategies have a minimal impact on accuracy for 5-shot settings, as demonstrations help regularize the output format. The difference is more evident in the 0-shot setting, likely related to the specific tokenizers. E.g., Llama3-8B treats the 2 characters " A" as 1 token, and has a slightly higher accuracy when using the 1st generated token as the answer. Conversely, Llama2-7B and Mistral-7B tokenizers treat "A" as 1 token. Using the option ID token with the highest probability as the answer for those models generally leads to higher accuracy because it disregards the generation probability of other irrelevant tokens, e.g., "\n", " ".

Table 8: Comparing two evaluation strategies for open-source LLMs: option ID token with maximum probability and first new token.

| Model | Eval | English MMLU | | Mixup MMLU | |
|---|---|---|---|---|---|
| | | Max prob | 1st token | Max prob | 1st token |
| **Llama2-7B** | 0-shot | **41.53** | 37.74 | **32.18** | 27.47 |
| | 5-shot | 45.88 | **45.90** | 36.92 | **36.96** |
| **Mistral-7B** | 0-shot | **60.21** | 58.41 | **47.86** | 42.29 |
| | 5-shot | **62.57** | 62.54 | **51.07** | 51.05 |
| **Llama3-8B** | 0-shot | 60.54 | **62.11** | 48.62 | **50.13** |
| | 5-shot | 65.00 | **65.39** | 51.65 | **52.01** |

**Crosslingual evaluation of additional models on MMLU knowledge**   We present additional results for Llama2-7B, Zamba-7B, and Mixtral-8x7B. Figure 14 shows the monolingual evaluation of LLMs on MMLU, fully translated for non-English languages. The models consistently achieve higher accuracy in English compared to other languages.

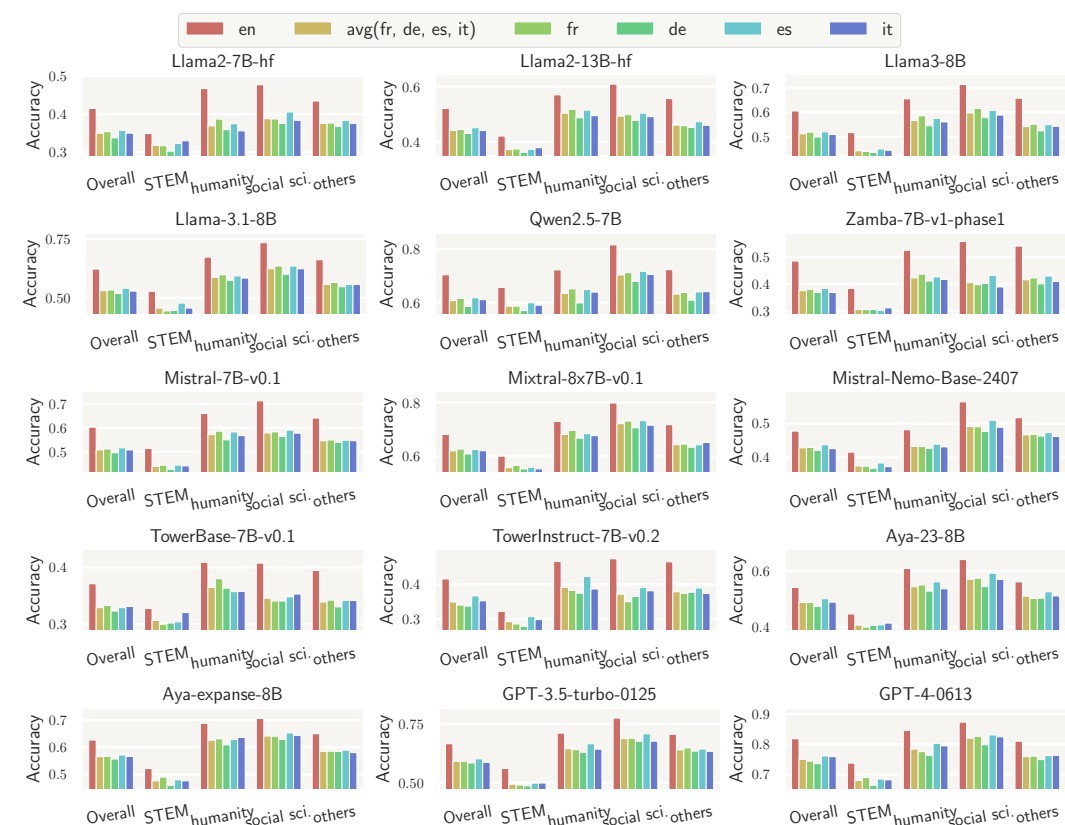

Figure 14: Monolingual evaluation of LLMs on MMLU (fully translated for non-English languages). LLMs consistently perform better at answering multi-choice questions in English than in other languages.

Figure 15 displays the accuracy of LLMs on MMLU variant benchmarks. We observe a significant drop in accuracy under crosslingual MCQ evaluation, especially for ground-truth translated MMLU variant, indicating cross-lingual knowledge barriers. The barrier is more pronounced in Llama2-7B and Zamba-7B than in Mixtral-8x7B, possibly due to the larger capacity and multilingual capabilities of Mixtral-8x7B.

### E.3 MIXED-LANGUAGE FINE-TUNING

**HP Quiz evaluation results of LLMs fine-tuned on WikiText-2** Fig. 8 in the main paper presents the Harry Potter Quiz evaluation results on Llama2-7B and Llama3-8B models fine-tuned on general knowledge corpora (i.e., WikiText-2). Fig. 16 presents additional results for the Llama2-13B (left) and Mistral 7B (right) models. (1) The trends are consistent with those observed for Llama2-7B and Llama3-8B, where fine-tuning on a mixed-language general corpus, WikiText-2, enhances the models' performance on the domain-specific HP Quiz task across multiple languages, including English. (2) Word-level language mixing is generally most effective for Llama2-13B, whereas sentence-level mixing is more effective for Mistral-7B.

**MMLU evaluation results of LLMs fine-tuned on WikiText-103** Tb. 2 in the main paper presents the English MMLU and mixup MMLU evaluation results on Llama2-7B and Llama3-8B models fine-tuned on general knowledge corpora (i.e., WikiText-103). Here we present additional results for Llama2-7B (Fig. 17) and Llama3-8B (Fig. 18) on more MMLU variant benchmarks, including full translation, question translation, options transition, and ground-truth option translation. We report the average accuracy (with *) across four non-English languages {fr, de, es, and it } for those settings.

(1) As shown in Fig. 17 and Fig. 18, models fine-tuned on mixed language WikiText-103 (whether at the word level or sentence level) generally achieve better performance than those fine-tuned on the original English WikiText-103 or the non-fine-tuned models, especially in the GT-option

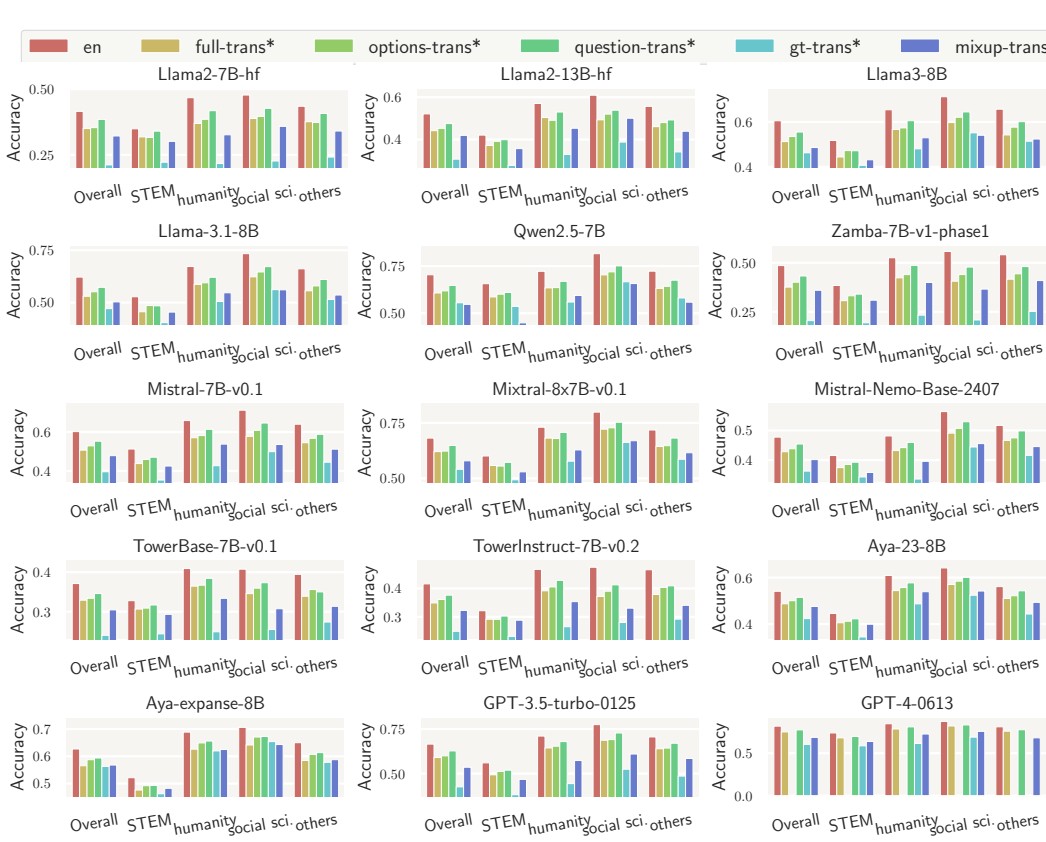

Figure 15: Crosslingual evaluation of LLMs on MMLU variant benchmarks. The bars with * denotes the average accuracy across {fr, de, es, and it}. LLMs perform better at answering MCQs in English than in mixed-language settings, especially the ground truth option and mixup translation, indicating the existence of cross-lingual knowledge barriers. Due to budget constraints, GPT-4 is evaluated only in the most challenging settings.

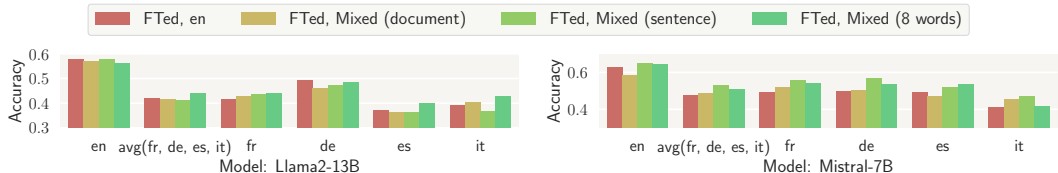

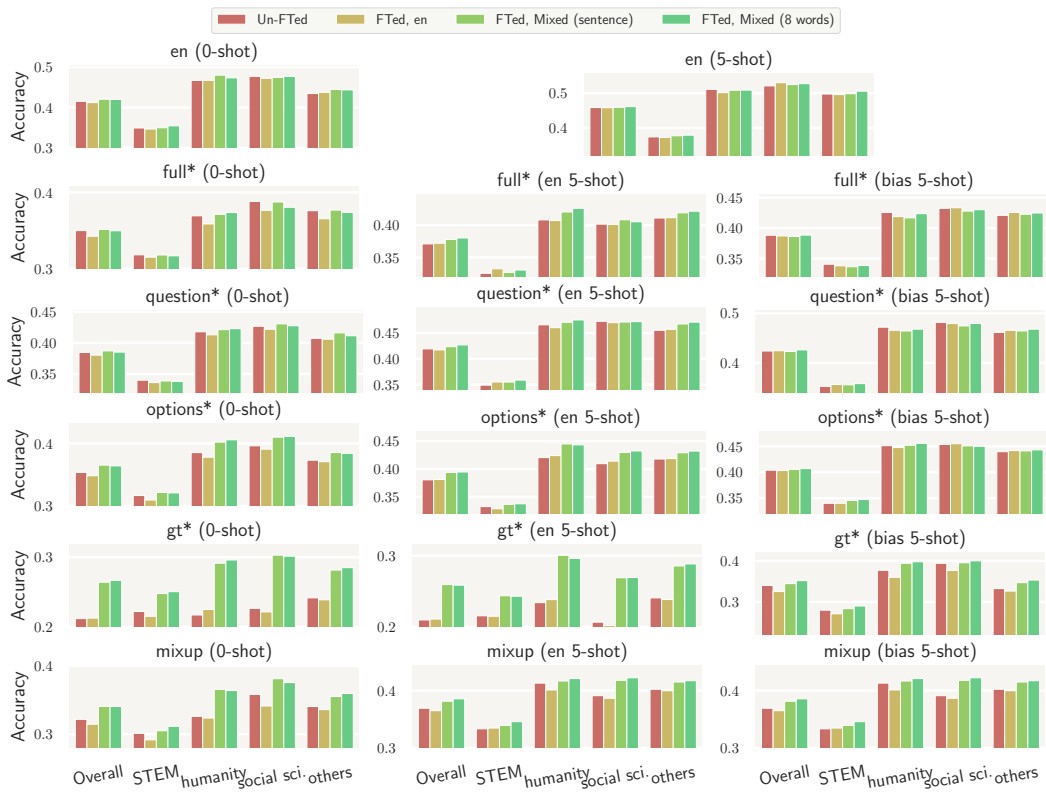

Figure 16: Fine-tuning on a mixed-language general corpus (e.g., WikiText-2) enhances the model's performance on domain-specific task (e.g., Harry Potter knowledge test) across multiple languages, including English.

Figure 17: Performance of Llama2-7B models on MMLU variant benchmarks. Fine-tuning on mixed language WikiText-103 generally outperforms fine-tuning on English WikiText-103 or using the non-fine-tuned model.

translated and mixup translated MMLU setups. These two evaluation setups originally had the lowest performance for the non-fine-tuned model, and thus the cross-lingual ability gains after fine-tuning are more apparent. These results suggest that multiple language switches during fine-tuning enable LLMs to better understand and process multilingual input and leverage cross-lingual knowledge for commonsense reasoning tasks. (2) An exception to this trend is observed with the GT-option translated MMLU under the 5-shot biased demonstrations setting for Llama3-8B, where performance drops. This drop is likely due to the non-fine-tuned Llama3-8B's stronger tendency to follow biased demonstrations, using a shortcut to select the non-English option as the answer. (3) Fine-tuning models on a mixed-languages corpus performs better than other models across different 0-shot and few-shot scenarios, particularly in the 0-shot setting and 5-shot English demonstrations setting. While 5-shot biased demonstrations lead to the best performance, they are less applicable than English demonstrations in real-world scenarios, as we cannot know in advance the language mixing pattern of user queries.

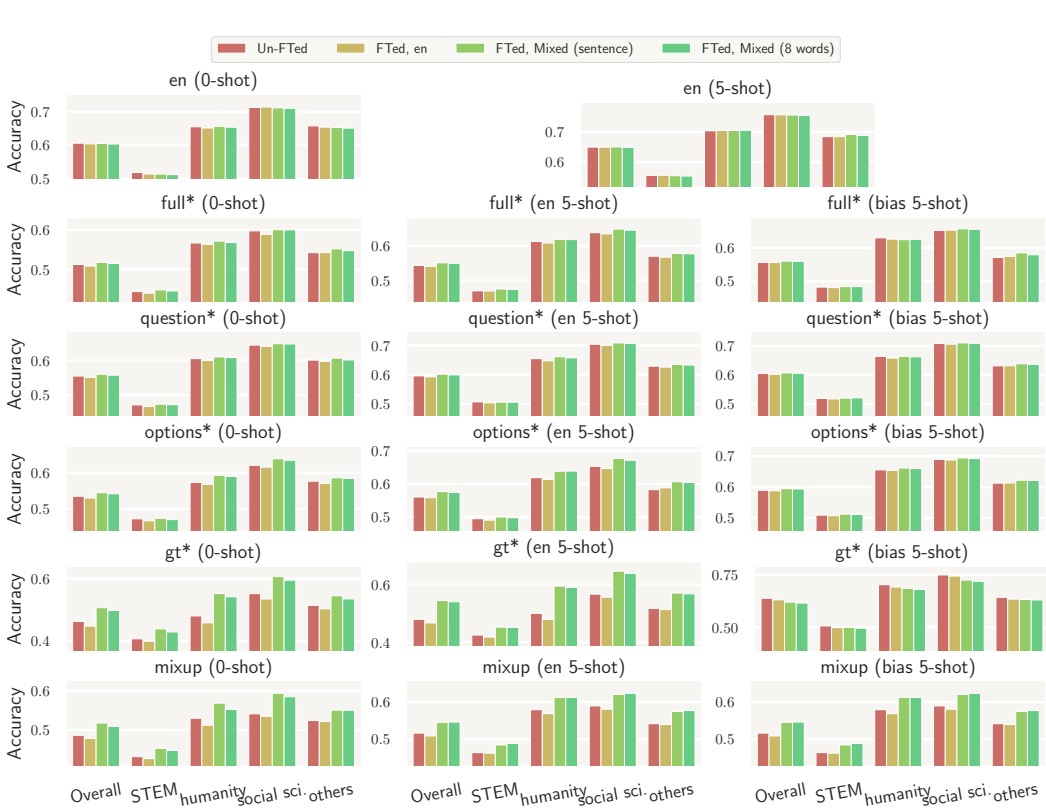

Figure 18: Performance of Llama3-8B models on MMLU variant benchmarks. Fine-tuning on mixed language WikiText-103 generally outperforms fine-tuning on English WikiText-103 or using the non-fine-tuned model.

