# OpenReview forum: "Crosslingual Capabilities and Knowledge Barriers in Multilingual Large Language Models"
_ICLR.cc/2025/Conference — Submitted to ICLR 2025_

### Official Review · Reviewer_uRPF · 2024-11-03

**Soundness:** 2
**Presentation:** 2
**Contribution:** 1
**Rating:** 3
**Confidence:** 4

**Summary:**

The paper claims to investigate the cross-lingual knowledge barrier for four high-resource non-English languages by examining benchmark score variability under different translation scenarios for six LLMs. However, it suffers from several weaknesses, including:

- A lack of detail regarding how sentence embeddings are obtained to demonstrate explicit cross-lingual capabilities. And no justification for the chosen probability settings.
- Uncontrolled experiments that do not sufficiently prove that the results are due to the cross-lingual knowledge barrier rather than the performance of the languages in non-knowledge related tasks.

In the rest of the paper, the authors evaluate well-established methods to improve the multilinguality score. Their proposal to construct a fine-tuning dataset comprising examples from multiple languages, particularly at the sentence level, aims to achieve better scores compared to English fine-tuning. However, I believe this outcome is completely expected and does not represent a significant contribution.

**Strengths:**

- The paper demonstrates that a cross-lingual knowledge barrier exists for four non-English languages through experiments evaluating how much benchmark scores vary when the benchmarks are translated under different scenarios.

- The authors assess well-established methods to improve the multilinguality score and propose constructing a fine-tuning dataset that comprises examples from multiple languages, particularly at the sentence level, to achieve better scores in multilinguality.

**Weaknesses:**

- L26-36: The first figure presents a poor example. Why couldn't it simply be the same question in both English and French? Additionally, why is there a need to create a knowledge that does not exist, so it can not be tested out as an example?

- L104: The authors state, “such presence, if it exists at all,” regarding parallel sentences on the web. It would be interesting to know how papers that mine billions of parallel texts from the web feel about this statement (e.g., https://arxiv.org/abs/1911.04944).

- L137: The paper mentions obtaining sentence embeddings from the LLM; however, it does not detail how this is done. Since these are decoder-only models with causal attention, how are the embeddings of the tokens aggregated? Is it the last token, or are they weighted? (see section 3.1 of https://arxiv.org/pdf/2410.05873 for a background on the different techniques)

- L134-142: There are many variables concerning probabilities, yet there is no explanation for why these specific values were chosen.

- L158-159: The statement “This underscores the explicit cross-lingual capabilities of multilingual LLMs” lacks clarity regarding the rationale and conclusions drawn. With a lower probability (0.16 vs. 0.8), tokens were randomly selected from the vocabulary. Now, the distribution of these two (cosine similarities between the original and mixed translated sentence embeddings versus the original and random token-replaced sentence embeddings) varies significantly. Many variables are involved here: do the random tokens from the vocabulary come only from all writing systems or just Latin? How are the embeddings computed, and why were these probabilities chosen? Although I acknowledge the existence of cross-lingual capabilities, I am not convinced by the authors' experiment.

- L199-200: There is no experiment/evaluation/audit demonstrating how valid these translations are.

- L247: Conclusion 1 does not necessarily indicate a knowledge transfer problem. In English-centric LLMs, it is expected that performance will be better in English. Simply stating that English performs better in an English-centric LLM does not imply that knowledge cannot be transferred; rather, it suggests that performance in the secondary language may be low, even in language understanding tasks. The authors selected languages where LLMs typically perform better (but is it on the level of English?), so it is not surprising that they cannot fully grasp the knowledge of English. If the authors had chosen low-resource languages, we might observe even lower performance, which could indicate that the LLM lacks substantial knowledge of those languages to begin with than other knowledges.

For the rest of the paper, the authors evaluate well-established methods to improve the multilinguality score. Their proposal to construct a fine-tuning dataset comprising examples from multiple languages, particularly at the sentence level and achieve better scores compared to English fine-tuning. However, I believe this outcome is completely expected and does not represent a significant contribution.

**Questions:**

See weaknesses part especially L137, L134-142, L158-159, L199-200.

---

> ### Author Response · Authors · 2024-11-23
> **Response to Reviewer uRPF  (Part 1)**
>
> Thank you for your feedback! We address your questions and comments below.
>
> > Q1:  L26-36: The first figure presents a poor example. Why couldn't it simply be the same question in both English and French? Additionally, why is there a need to create a knowledge that does not exist, so it can not be tested out as an example?
>
> Thank you for the valuable comment. Following your suggestion, we have updated Figure 1 to use the same question in both English and French.
>
> Regarding the use of a fabricated example, it was purely for illustration purposes. The goal was to demonstrate that the model must rely on knowledge stored in its weights in one language to answer questions in another language. Since the exact information on which knowledge is present in which language in the training data of existing pre-trained LLMs is not fully transparent, we opted to use a fabricated example to avoid potential confounding factors.
>
>
> > Q2:  L104: The authors state, “such presence, if it exists at all,” regarding parallel sentences on the web. It would be interesting to know how papers that mine billions of parallel texts from the web feel about this statement (e.g., https://arxiv.org/abs/1911.04944).
>
> Thank you to the reviewer for pointing this out. We acknowledge the potential existence of parallel data in the training data of those LLMs, which is unknown to us. We revised our statements in the Introduction (Section 1) to clarify this point and have incorporated a discussion with [1] shared by the reviewer.
>
> Reference:
> - [1] CCMatrix: Mining Billions of High-Quality Parallel Sentences on the WEB. https://arxiv.org/abs/1911.04944
>
> > Q3:  L137: The paper mentions obtaining sentence embeddings from the LLM; however, it does not detail how this is done. Since these are decoder-only models with causal attention, how are the embeddings of the tokens aggregated? Is it the last token, or are they weighted? (see section 3.1 of https://arxiv.org/pdf/2410.05873 for a background on the different techniques)
>
> Thanks for the question. The sentence embedding is a single vector representing the average of the last layer's activations across all tokens in the sentence,  a common pooling strategy for creating fixed-size representations of variable-length text.  We have added these clarifications to the revised PDF.
>
> > Q4:  L134-142: There are many variables concerning probabilities, yet there is no explanation for why these specific values were chosen.
>
>  Thanks for raising this point.
> - For Mixed-language-translate, the choice of a probability of $p = 0.8$ that each word is unchanged corresponds to a 20% probability ($1 - p = 0.2$) that each word is replaced. This aligns with the uniform probability distribution across the five main languages evaluated in our study: {en, fr, de, es, it}.  In other words, each word has a $0.16$ probability of being translated into a non-English language.
> - This also explains our choice for Random Token Replacement where $p=0.16$.
>
> We have added these clarifications to the revised PDF.
>
> > Q5:  L158-159: The statement “This underscores the explicit cross-lingual capabilities of multilingual LLMs” lacks clarity regarding the rationale and conclusions drawn. With a lower probability (0.16 vs. 0.8), tokens were randomly selected from the vocabulary. Now, the distribution of these two (cosine similarities between the original and mixed translated sentence embeddings versus the original and random token-replaced sentence embeddings) varies significantly. Many variables are involved here: do the random tokens from the vocabulary come only from all writing systems or just Latin? How are the embeddings computed, and why were these probabilities chosen? Although I acknowledge the existence of cross-lingual capabilities, I am not convinced by the authors' experiment.
>
>
> Thanks for the insightful comment.  The random tokens come from the entire vocabulary space of the tokenizer.   The embedding computation is clarified in answer to Q3.
> We kept the token modification probability low (0.16) across all three variants: mixed-language translation, random token replacement, and random token dropout, to ensure that the altered sentences could maintain semantic and structural similarity with the original text.

---

> ### Author Response · Authors · 2024-11-23
> **Response to Reviewer uRPF (Part 2)**
>
> > Q6:  L199-200: There is no experiment/evaluation/audit demonstrating how valid these translations are.
>
> Thank you for the comment. We utilized the commercial system Google Translate API to perform the translations, which is regarded in the literature as the strongest baseline for machine translation tasks [1].
>
> However, we acknowledge that even state-of-the-art translation tools may not perfectly capture nuanced or domain-specific meanings in all languages. As the authors are not fluent in all the languages evaluated, it is challenging to independently verify the accuracy of the translations across different foreign languages. We recognize this limitation and have explicitly noted it in the limitations section (Appendix A) of the revised PDF.
>
> Reference:
> - [1] Wenhao Zhu, Hongyi Liu, Qingxiu Dong, Jingjing Xu, Shujian Huang, Lingpeng Kong, Jiajun Chen, and Lei Li. Multilingual machine translation with large language models: Empirical results and analysis. In NAACL, 2024.
>
>
> > Q7:   L247: Conclusion 1 does not necessarily indicate a knowledge transfer problem. In English-centric LLMs, it is expected that performance will be better in English. Simply stating that English performs better in an English-centric LLM does not imply that knowledge cannot be transferred; rather, it suggests that performance in the secondary language may be low, even in language understanding tasks. The authors selected languages where LLMs typically perform better (but is it on the level of English?), so it is not surprising that they cannot fully grasp the knowledge of English. If the authors had chosen low-resource languages, we might observe even lower performance, which could indicate that the LLM lacks substantial knowledge of those languages to begin with than other knowledges.
>
> Thank you for the comment. We would like to clarify that **Conclusion 1** is based on **traditional monolingual** evaluations, not our proposed cross-lingual evaluations. As noted in Section 3.1 (lines 174-184), *monolingual evaluation is inadequate for assessing crosslingual abilities* because it does not isolate the process of transferring knowledge across languages. Recognizing these limitations and confounding factors in previous multilingual evaluations, we designed controlled setups to study cross-lingual transfer.
> Specifically, we proposed the Mixed-Language Evaluation strategies (lines 185-197), which directly invoke cross-lingual capabilities by incorporating novel compositions of multiple languages in multi-choice QA formats. The conclusions derived from **proposed cross-lingual evaluation** strategies are presented in **Conclusions 2–4** and **Evaluation on 16 languages** (lines 248–297). We apologize for any confusion caused by the presentation of results.
>
> Moreover,  we investigate actual cross-lingual knowledge transfer via a controlled experimental setup in the Harry Potter Quiz dataset (Section 3.2). We fine-tune LLMs in in-domain English content and observe improved performance in other languages, indicating that the model is capable of transferring knowledge across languages. However, the gap remains between English and other languages, indicating the cross-lingual knowledge barriers: LLMs struggle to fully utilize the parametric knowledge acquired during English fine-tuning to answer related questions in other languages.  Therefore, in Section 4.2, we proposed mixed-language fine-tuning methods, to achieve improved cross-lingual performance, which further supports that such knowledge gaps can be narrowed with more effective cross-lingual knowledge transfer methods.
>
> We hope this clarification addresses the reviewer's concerns.

---

> ### Author Response · Authors · 2024-11-23
> **Response to Reviewer uRPF (Part 3)**
>
> > Q8  For the rest of the paper, the authors evaluate well-established methods to improve the multilinguality score. Their proposal to construct a fine-tuning dataset comprising examples from multiple languages, particularly at the sentence level and achieve better scores compared to English fine-tuning. However, I believe this outcome is completely expected and does not represent a significant contribution.
>
> Thank you for the comment. While the outcome of mixed-language fine-tuning improving multilinguality scores may seem intuitive, we study their effectiveness on different mixing units (words, sentences, documents). We also would like to emphasize the broader contributions and novelty of our work beyond this specific result:
>
> 1. **New Problem Formulation**: We explicitly formulate the problem of the crosslingual knowledge barrier in pretrained/finetuned language models, focusing on their ability to retrieve and utilize parametric knowledge across languages. This goes beyond surface-level multilinguality assessments and addresses deeper challenges in crosslingual knowledge transfer.
>
>
> 2. **Proposed Challenging Scenarios**: We introduce novel crosslingual QA scenarios, including multi-choice QA tasks with innovative compositions of multiple languages, to specifically evaluate the models’ implicit crosslingual knowledge capabilities in general (MMLU benchmark) and domain-specific (Harry Potter quiz) contexts.
>
>
> 3. **Revealing Crosslingual Knowledge Barriers**: We evaluate a total of 15 multilingual LLMs and 16 languages (We have added 11 languages and 9 LLMs during the rebuttal phrase following the reviewers’ suggestions in the revised PDF). Our results show that while models may demonstrate promising surface-level cross-lingual capabilities (e.g., translation and embedding space analyses), they struggle with deeper cross-lingual knowledge transfer. This finding reveals a previously underexplored limitation in existing multilingual LLMs.
>
>
> 4. **Limited Impact of Inference-Time Mitigations**: Our study highlights that simple inference-time mitigation methods provide limited/no improvement, further emphasizing the challenge posed by the crosslingual knowledge barrier.
>
>
> 5. **Generalization of Mixed-Language Fine-Tuning**: We proposed a mixed-language fine-tuning method across different mixing units (words, sentences, documents). The results on the MMLU variant and Harry Potter quiz show that our proposed method not only improves crosslingual performance for the languages included during fine-tuning but also benefits languages that were unseen during fine-tuning, including both low-resource languages and languages with different token distributions. This observation highlights the broader applicability of our approach and its impact on underrepresented languages.
>
>
> We believe these contributions collectively advance the understanding of cross-lingual knowledge transfer in LLMs and provide a meaningful foundation for future research in this area.

---

> ### Author Response · Authors · 2024-11-27
> **Follow-up**
>
> Dear Reviewer,
>
> Thank you once again for your detailed comments and suggestions. As the revision period is approaching its end, we would greatly appreciate your feedback on whether our responses and revision have addressed your concerns. We are also happy to engage in further discussions if needed.

---

> ### Author Response · Authors · 2024-12-02
> **Follow-up with Reviewer uRPF**
>
> We wanted to express our gratitude once again to the reviewer for the feedback. We hope that the detailed responses and the manuscript revisions we provided have clarified the concerns regarding Figure 1, parallel data, hyperparameter details, the distinction between monolingual evaluation and our cross-lingual knowledge reasoning evaluation, as well as our contributions beyond fine-tuning.
>
> In response to the reviewers’ comments, we have expanded our evaluation to include a total of 16 languages and 15 multilingual LLMs to provide broader insights into our study. Our expanded evaluation covers low-resource languages and languages with token distributions significantly different from English, as well as state-of-the-art multilingual LLMs such as Qwen2.5-7B, Llama-3.1-8B, Mixtral-8x7B-v0.1, aya-expanse-8B, aya-23-8B, Mistral-Nemo-Base-2407, Tower-7B-base and Tower7B-instruction-tuned model.
>
> If there are any remaining concerns or if further clarification is needed, we would be more than happy to address them before the discussion period concludes. We also kindly invite you to reevaluate our work in light of the additional clarifications and improvements we have presented.
>
> Thank you again for your time and feedback.

---

> > ### Comment · Reviewer_uRPF · 2024-12-02
> >
> > Thank you for your responses. I still do not think this paper has enough contributions. The major drawbacks for me are that the embeddings are not calculated in an appropriate way for decode-only models, the experiments are not controlled enough to necessarily indicate a knowledge transfer problem, and all the methods used to improve the multilinguality score are well-established and not a contribution. I maintain my score (3).

---

> ### Author Response · Authors · 2024-12-03
> **Response to Reviewer uRPF (part 1)**
>
> Thank you for your feedback. We respectfully believe that the questions raised may stem from misunderstandings that we are happy to clarify further. Below, we address each of your comments in detail:
>
>
> >  embeddings are not calculated in an appropriate way for decode-only models,
>
> We respectfully disagree with this comment.  Mean pooling of last hidden states is a **standard approach** to calculate sentence embedding specifically for **decoder-only** models. This method has been employed in widely-adopted open-source libraries, such as llama.cpp (a inference service of Llama-series models with over 68,000 GitHub stars) [1], and has been used in recent research publications [2,3].
>
> We are open to exploring alternative embedding methods - If you could suggest specific techniques, we are happy to also report them in Section 2.
>
> Additionally, we would like to note that the paper [4], which the reviewer cited in the original **Q3**, is not published in a peer-reviewed venue, which raises questions about its appropriateness.
>
>
> Reference
> - [1] https://github.com/ggerganov/llama.cpp/issues/6754
> - [2] LLMEmbed: Rethinking Lightweight LLM's Genuine Function in Text Classification https://arxiv.org/abs/2406.03725, ACL 2024
> - [3] LLM2Vec: Large Language Models Are Secretly Powerful Text Encoders https://arxiv.org/abs/2404.05961, COLM 2024
> - [4] MEXA: Multilingual Evaluation of English-Centric LLMs via Cross-Lingual Alignment  https://arxiv.org/pdf/2410.05873
>
>
>
> > the experiments are not controlled enough to necessarily indicate a knowledge transfer problem
>
> Could the reviewer please clarify which specific experiments are considered "not controlled enough"? This clarification would help us better understand and address the concern.
>
> If the reviewer is referring to **Conclusion 1** as mentioned in the original **Q4**, we would like to reiterate that this conclusion is based on traditional monolingual evaluations, which were **not** the primary focus of our study. Instead, our key findings are derived from the proposed cross-lingual evaluation strategies,  which are presented separately in **Conclusions 2–4 (lines 248–297) and are supported by results across 16 languages**, which provide evidence of a **crosslingual knowledge transfer problem**.
>
> Specifically, we adopt an inherent crosslingual interaction approach by introducing mixed-language multiple-choice question formats. These formats are purposefully designed as **novel cross-lingual compositions** that are unlikely to have been encountered during pretraining. This ensures that our experiments evaluate genuine crosslingual knowledge transfer rather than reliance on each single language.
>
> If the reviewer identifies particular experiments that are "not controlled enough", we would greatly appreciate a detailed explanation of the perceived shortcomings.

---

> ### Author Response · Authors · 2024-12-03
> **Response to Reviewer uRPF (part 2)**
>
> > all the methods used to improve the multilinguality score are well-established and not a contribution
>
> We respectfully disagree with this comment and request clarification regarding which methods the reviewer considers "well-established." Below, we highlight the novelty of our work and provide evidence that our contributions are original and impactful:
>
> The proposed mixed-language fine-tuning (varying the mixing unit: words, sentences, documents) is, to the best of our knowledge, novel in the context of multilingual LLMs.  This approach provides new insights into cross-lingual knowledge reasoning:
> - Mixed-language fine-tuning boosts cross-lingual capabilities (line 421-429): By exposure to frequent language switch during fine-tuning on general (out-of-domain) corpus, LLMs can better adapt to the setting during testing when the same knowledge is asked in a different (and usually non-English) language.
> - Generalization to out-of-distribution languages (lines 492-512):  We evaluated our fine-tuned models on languages that were not included in fine-tuning data. Results in Fig. 11 show that mixed-language fine-tuning on the general Wiki corpus can improve the performance of 11 other languages on HP-Quiz, including low-resource ones and those substantially different from English. Furthermore, as shown in Fig. 12, mixed-language fine-tuning also boosts the performance of MMLU variants in various cross-lingual settings for four low-resource languages.
>
> As detailed in our response to the original **Q8**, our work also makes several broader contributions to the field, including:
> - New Problem Formulation: Identifying and addressing the cross-lingual knowledge barrier in pretrained and fine-tuned language models.
> - Proposed Challenging Scenarios and Datasets: Specifically designed to evaluate cross-lingual knowledge reasoning.
> - Revealing Cross-Lingual Knowledge Barriers: A detailed analysis of 15 multilingual LLMs across 16 languages.
> - Limited Impact of Inference-Time Mitigations: Our work provides concrete evidence that inference-time strategies fail to address these challenges.
> - Generalization of Mixed-Language Fine-Tuning: Comprehensive evaluations demonstrate the robustness of our proposed approach.
>
> If the reviewer perceives our methods to be "well-established", we respectfully request supporting references. Such information would help us better understand this perspective and either further justify the novelty of our contributions or make adjustments as necessary to address these concerns.
>
> Thank you for your time and consideration.

---

### Official Review · Reviewer_2t3Z · 2024-11-04

**Soundness:** 1
**Presentation:** 3
**Contribution:** 1
**Rating:** 6
**Confidence:** 4

**Summary:**

Authors study multilingual abilities and extent of crosslingual transfer of decoder transformer language models, using three different methodologies: embedding-based, MMLU based and domain-specific. In embedding part authors collect embeddings of sentences in English, sentences, in which some of the input tokens were corrupted by masking the attention or randomly replacing them with different tokens from vocabulary of the model and sentences, in which words were randomly translated into a different language and calculate distances between English, corrupted and partially translated languages. General knowledge based evaluation is conducted by translating parts of the questions or answers from MMLU dataset into a different language. In domain-specific evaluation authors ask questions about Harry Potter series and apply same translation techniques as in general knowledge evaluation. First evaluation show that models have close embeddings after per token translations. Second evaluation shows that models are doing much worse on MMLU after partial translations of the answers. Third evaluation shows that applying translation to parts of the questions or answers from Harry Potter trivia the crosslingual barrier still exists. In later parts of the paper authors test different methodologies of overcoming this barrier, either by prompt engineering and doing few-shot prompting which has limited success and by finetuning selected models on WikiText-2 for general text knowledge and WikiText-103 for domain specific trivia, which improves the performance on both MMLU and HP-Quiz.

**Strengths:**

1. Experiments with embeddings of the models are interesting and give some insight into how models encode meaning of partially translated sentences.

2. Experiments with general purpose knowledge can be used to better understand multilingual capabilities of trained models.

3. Mixup-version of MMLU, when provided, will be useful to do better multilingual evaluations, since the language perturbations can successfully combat the test leakage.

4. The text is clear and understandable.

**Weaknesses:**

1. Models, selected on line 100, are not multilingual. Llama-3-8B was explicitly stated as a monolingual model, focusing only on English, with usage in other languages described as out-of-scope use. Multilinguality was only stated in 3.1’s model card. Same can be said about Llama-2-7B and Llama-2-13B. Mistral-7B is not explicitly stated to be multilingual, either. Given both multilingual evaluations and community reports, it is safe to say that these models do not exhibit strong multilingual capabilities. Authors additionally evaluate Mixtral 8x7B and Aya-23-8b, which are explicitly stated to be multilingual models and have some knowledge in selected languages (en, fr, de, es, it), but they are mostly ignored in the main part of paper, instead focusing on Llama-2 and Llama-3. To my mind this is a big misstep, since bad performance of the selected models could be incorrectly generalized to truly multilingual models.

2. Domain specific evaluations are being done on the Harry Potter series of books, which was translated to different languages with varying quality. This raises the question of how well the terms described in the books are translated: for instance, tha name of Neville Longbottom is translated to Italian as Neville Paciock, which retains the connotation of clumsiness, but does not sound similarly and does not preserve the meaning of long bottom with the English version, to Chinese as Nàwēi Lóngbādùn, which does not retain the connotation since it’s a simple transliteration and to Russian as Nevil Dolgopups, which both does not retain the clumsiness connotation, meaning or common sound/tokens. This raises concerns if the arguments presented in the paper (e.g. knowledge learned in one language can be translated to other languages) can be checked via Harry Potter trivia.

3. All of the selected models almost certainly have seen the original books, dumps from HP Wiki or Wikipedia and blog posts about Harry Potter in non-English languages, so it may have leaked into the training data for all of the languages.

4. It is common knowledge that both instruction and preference tuning causes catastrophic forgetting, which leads to models being better at instruction following, safety and answer formatting, but worse at parametric knowledge. From the list of the described models we can see that only Llama-2, Llama-3, Mixtral-8x7b, Zamba-7b and Mistral-7b models are available as base models, with GPT-3.5, GPT-4, Aya-23 being instruction tuned models. It is not explicitly stated, which versions of the models are used – pretrained or finetuned, Thus, some clarification is required, since it directly influences the amount of multilingual capability of the models. If these are instruction tuned models, further finetune of them on Harry Potter dataset could lead to catastrophic forgetting, thus perturbating the models’ scores and if these are base models, direct comparison between Aya-23 and llamas, for instance, presents a potential inconsistency in methodology.

5. The phenomenon of crosslingual knowledge barrier (e.g. limited multilinguality capabilities of the model) is well known, so the main contribution of the paper lacks originality.

**Questions:**

1. It would be very beneficial if the authors of the paper would use different languages, such as Chinese, Hebrew, Russian or Hindi, since these languages have a much lower amount of shared tokens with English, which is selected as the most highly resourced language in the training mix of all selected models. This would give us insight into how the applied methodology translates into languages, which the model has a harder time training due to lesser overlap between dictionaries.

2. Please, repeat your evaluations using a different set of models, such as Mistral-Nemo-12B, Qwen-2.5-7B, Command-R and LLaMA-3.1-8B which are multilingual by design. Claims of multilinguality of these models are supported by extensive multilingual evaluations both by model authors and community members.

3. Perhaps designing another evaluation dataset for domain specific knowledge, based on something like some local company guidelines or university campus rules and translating it into different languages would make a much stronger point to support the claims, since it would not exhibit translation artifacts.

---

> ### Author Response · Authors · 2024-11-23
> **Response to Reviewer 2t3Z (Part 1)**
>
> Thank you for your feedback! We address your questions and comments below.
>
> > Q1: Models, selected on line 100, are not multilingual. Llama-3-8B was explicitly stated as a monolingual model, focusing only on English, with usage in other languages described as out-of-scope use.... To my mind this is a big misstep, since bad performance of the selected models could be incorrectly generalized to truly multilingual models…… Please, repeat your evaluations using a different set of models, such as Mistral-Nemo-12B, Qwen-2.5-7B, Command-R and LLaMA-3.1-8B which are multilingual by design. Claims of multilinguality of these models are supported by extensive multilingual evaluations both by model authors and community members.
>
> Thank you for the thoughtful comment. Following the reviewer's suggestion, we have expanded our evaluation to *include a total of 15 multilingual LLMs in our main paper* in revised PDF, including strong multilingual LLMs such as Qwen2.5-7B, Llama-3.1-8B, Mistral-Nemo, Mixtral-8x7B-v0.1, aya-expanse-8B, aya-23-8B and  Tower-series 7B models.
>
> While our original focus -- Llama-3-8B, Llama-2-7B, Llama-2-13B, and Mistral-7B -- are not explicitly stated to be multilingual, we find that they have *competitive cross-lingual performance* to LLMs that are explicitly mentioned to be multilingual.
> - From the results on cross-lingual MMLU tasks in  Figure 4, we find that the multi-choice question answering accuracy of  Llama-3-8B and Llama-3.1-8B are very close, potentially due to their similar knowledge scope in training data. Moreover, Mistral-7B performs better than aya-23-8B, and Llama-2-13b performs better than Mistral-Nemo.
> - From the results of the Harry Potter Quiz tasks in Figure 7, we find that Mistral-7B and Llama-3-8B obtain comparable performance to other state-of-the-art multilingual LLMs.
>
> These new results *justify our original model selection*. Moreover, we clarify that our primary goal was to evaluate general-purpose LLMs that can understand multiple languages due to massive pretraining data as stated in Section 1. While models like Llama-3-8B, Llama-2-7B, Llama-2-13B, and Mistral-7B may not be explicitly optimized for multilingual tasks, it is reasonable to assume that training data in non-English Languages exists in their web-crawled pretraining datasets. Our translation results in Section 2.1 (Table 7 in the Appendix) show that these models achieve competitive performance to commercial-level translation API in explicit translation tasks, particularly in translations from other languages to English. Additionally, the embedding analysis in Section 2.2 shows the explicit cross-lingual capabilities of those LLMs. These results motivated us to consider them valid multilingual LLMs for our study.
>
> Overall,  the results in Figure 4 and Figure 7 show that the 15 multilingual LLMs demonstrate significant cross-lingual knowledge barriers on our tasks, suggesting that such limitations are not exclusive to less multilingual-optimized models, and generalize to strong multilingual models.
>
> > Q2: Domain specific evaluations are being done on the Harry Potter series of books, which was translated to different languages with varying quality. This raises the question of how well the terms described in the books are translated: for instance,...., which both does not retain the clumsiness connotation, meaning or common sound/tokens. This raises concerns if the arguments presented in the paper (e.g. knowledge learned in one language can be translated to other languages) can be checked via Harry Potter trivia.
>
> Thanks for the insightful comment!  We acknowledge that translation is a very challenging task and it is difficult even for expert human translators to get all the subtlety right.
>
> However, we believe these challenges do not invalidate the motivation of our study.
> - For proper nouns, such as names, there is typically a finite set of mappings between languages (e.g., “Neville” to “Nàwēi” in Chinese or “Nevil” in Russian). While these mappings may not fully preserve connotations or meanings, a multilingual human with knowledge of both languages would still be able to transfer the relevant knowledge about the character or concept and answer questions accurately, even when the translation quality varies. Our work seeks to assess whether LLMs exhibit a similar capability.
> - More importantly, our focus on *multiple-choice QA tasks narrows down the search space for the correct answer*. Even with imperfect translations, a model with the relevant knowledge can identify the correct choice from a limited set of options. This approach mitigates some of the noise introduced by translation variability and allows us to measure whether the model truly possesses crosslingual knowledge.
>
> We appreciate your concern and have included the discussion of the limitations arising from translation variability in Appendix A of the revised PDF.

---

> ### Author Response · Authors · 2024-11-23
> **Response to Reviewer 2t3Z (Part 2)**
>
> > Q3  All of the selected models almost certainly have seen the original books, dumps from HP Wiki or Wikipedia and blog posts about Harry Potter in non-English languages, so it may have leaked into the training data for all of the languages.
>
> Thank you for the comment. Unfortunately, we cannot verify the extent to which the original books, Harry Potter Wiki dumps, or related content in non-English languages might have been included in the pretraining data for the selected models. This could vary significantly across languages and models. However, we note that the mere presence of such content in the training data does not guarantee that the model has learned this knowledge in a usable form, particularly across different languages. The ability to effectively utilize this knowledge to solve relevant question-answering tasks involves more than just observing those texts during training; it requires the model to generalize and reason based on its learned representations.
>
> Moreover, in Figure 6,  we conducted a controlled experiment where we *explicitly* finetuned the models on in-domain content presented in English, where models still consistently perform better at answering questions in English than in other languages. This suggests that LLMs struggle to fully utilize the parametric knowledge acquired during English fine-tuning to answer related questions in other languages and close the performance gap, indicating the presence of cross-lingual knowledge barriers.
>
> > Q4:  It is common knowledge that both instruction and preference tuning causes catastrophic forgetting, which leads to models being better at instruction following, safety and answer formatting, but worse at parametric knowledge. From the list of the described models we can see that only Llama-2, Llama-3, Mixtral-8x7b, Zamba-7b and Mistral-7b models are available as base models, with GPT-3.5, GPT-4, Aya-23 being instruction tuned models. It is not explicitly stated, which versions of the models are used – pretrained or finetuned, Thus, some clarification is required, since it directly influences the amount of multilingual capability of the models. If these are instruction tuned models, further finetune of them on Harry Potter dataset could lead to catastrophic forgetting, thus perturbating the models’ scores and if these are base models, direct comparison between Aya-23 and llamas, for instance, presents a potential inconsistency in methodology.
>
> Thank you for your insightful comments.  Following your suggestions, we have now clarified the used model versions in the updated figures of the revised PDF.
>
> As per the reviewer's suggestion, in our experiments, we fine-tuned the base models on Harry Potter-related content, avoiding instruction-tuned models, to mitigate the risk of catastrophic forgetting. Our primary focus in Section 4.2 was to compare the performance of the same model before and after fine-tuning to evaluate the effectiveness of our proposed mitigation method. This intra-model comparison avoids direct comparisons between different models (e.g., Aya-23 and Llama), ensuring consistency in the methodology and the conclusions drawn in Section 4.2.
>
> > Q5:  The phenomenon of crosslingual knowledge barrier (e.g. limited multilinguality capabilities of the model) is well known, so the main contribution of the paper lacks originality.
>
> Thank you for the comment. We clarify that the "limited multilinguality capabilities" often discussed in the literature are distinct from the "crosslingual knowledge barrier" phenomenon we study in this paper. While prior work primarily focuses on **explicit** cross-lingual tasks (e.g., machine translation) or **monolingua**l evaluation of benchmarks in each language, as seen in those model cards, our work introduces a *new challenge regarding cross-lingual knowledge reasoning*.
>
> Specifically, we evaluate models' capacity to **implicitly** retrieve and utilize parametric knowledge stored in their weights across languages to solve QA tasks – for both general knowledge (MMLU) and domain-specific knowledge (Harry Potter quiz). This is a different and underexplored aspect of **cross-lingual** performance that goes beyond tasks requiring direct language-to-language mappings (where the source text is provided in the context, so it is already "grounded") or simple monolingual evaluations (which do not necessarily invoke cross-lingual knowledge reasoning).
>
> Thank you again for raising this point and we hope our clarifications address the reviewer’s concerns.
>
>
> .

---

> ### Author Response · Authors · 2024-11-23
> **Response to Reviewer 2t3Z (Part 3)**
>
> > Q6: It would be very beneficial if the authors of the paper would use different languages, such as Chinese, Hebrew, Russian or Hindi, since these languages have a much lower amount of shared tokens with English, which is selected as the most highly resourced language in the training mix of all selected models. This would give us insight into how the applied methodology translates into languages, which the model has a harder time training due to lesser overlap between dictionaries.
>
> Thank you for the thoughtful comment.
> Following the reviewer's suggestion, we have expanded our evaluation to *include a total of 16 languages* and 15 multilingual LLMs to provide broader insights into our study.
> Specifically, in addition to the 2 germanic languages (English and German) and three romance languages (French, Italian, Spanish) we originally evaluated, we further consider
> - *Low-resource languages*: Malay (ms), Danish (da), Finnish (fi), Norwegian (no), Bengali (bn), Amharic (am);
> - *Languages with token distributions significantly different from English*: Russian (ru), Chinese (zn), Hebrew (he), Arbic (ar) and Hindi (hi).
>
> Conclusions:
> -  The results on MMLU (Figure 4 and Figure 5) and the Harry Potter quiz (Figure 7 ) show that cross-lingual knowledge barriers hold for those additional models and languages, which demonstrates the universality of our findings.
> - We also highlight that in Section 4.2, we evaluated our mixed-language fine-tuned LLMs on languages that were not used during the finetuning stage.  Our results in Figures 10 and 11  demonstrate that the model fine-tuned using our method on high-resource languages { English (en), French (fr), German (de), Spanish (es), and Italian (it)} can improve crosslingual performance on Harry potter quiz and MMLU for low-resource languages and languages that are rather different from English. These results provide encouraging evidence for the generalizability of our fine-tuning approach.
>
>
>
> > Q7:  Perhaps designing another evaluation dataset for domain specific knowledge, based on something like some local company guidelines or university campus rules and translating it into different languages would make a much stronger point to support the claims, since it would not exhibit translation artifacts
>
> Thanks for the great suggestion!   We greatly appreciate the idea of designing a domain-specific dataset based on local company guidelines or university campus rules. We also hope our response to Q3 has addressed the reviewer’s concern regarding the existing Harry Potter dataset, particularly as we have used in-domain fine-tuning to explicitly inject relevant English knowledge and evaluate cross-lingual knowledge barriers.
>
> Due to the limited time frame of the rebuttal phase and the experimental workload involved in evaluating additional models and languages, we have not yet completed the creation and evaluation of such a new dataset. We are committed to working on this new dataset based on your suggestion and will include it in our future revisions to further support our findings.

---

> ### Author Response · Authors · 2024-11-27
> **Follow-up**
>
> Dear Reviewer,
>
> Thank you once again for your detailed comments and suggestions. As the revision period is approaching its end, we would greatly appreciate your feedback on whether our responses and revision have addressed your concerns. We are also happy to engage in further discussions if needed.

---

> > ### Comment · Reviewer_2t3Z · 2024-11-29
> > **Response to the author's response**
> >
> > Dear Authors,
> >
> > Thank you for your detailed response and the thoughtful improvements made to your paper. I raised my score.
> >
> > >Thank you for the thoughtful comment. Following the reviewer's suggestion, we have expanded our evaluation to include a total of 15 multilingual LLMs in our main paper in revised PDF, including strong multilingual LLMs such as Qwen2.5-7B, Llama-3.1-8B, Mistral-Nemo, Mixtral-8x7B-v0.1, aya-expanse-8B, aya-23-8B and Tower series 7B models
> >
> > This enhancement significantly strengthens your evaluation and adds value to the overall study. I appreciate the effort in expanding the scope of your analysis.
> >
> > >While these mappings may not fully preserve connotations or meanings, a multilingual human with knowledge of both languages would still be able to transfer the relevant knowledge about the character or concept and answer questions accurately, evenwhen the translation quality varies
> >
> > I believe this claim still might need further qualification. While the general assertion that a multilingual human could transfer relevant knowledge is plausible, there are notable exceptions. Some translations differ so significantly that even speakers proficient in both languages may find it challenging to identify connections. For instance, one Russian translation of "Harry Potter" alters the character Severus Snape's name to "Zloteus Zley" (translated as "Evilus Evil"), Professor Quirrell becomes "Professor Strauns," Ravenclaw is translated to "Vransor," and horcruxes and boggarts are rendered as "okayant" and "vrizraks," respectively. Such substantial differences can obscure the original connotations, potentially invalidating the claim about consistent cross-language understanding. I conducted a simple experiment by asking my Russian-speaking colleagues, who had read all the books in both Russian and English, about these translated characters. They were either unable to identify the characters I referred to, or inferred them incorrectly. This validates my concerns about the translation quality.
> >
> > >Thank you for the comment. Unfortunately, we cannot verify the extent to which the original books, Harry Potter Wiki dumps, or related content in non-English languages might have been included in the pretraining data for the selected models. This could vary significantly across languages and models.
> >
> >  I must emphasize that this aspect raises critical concerns about the validity of your experiments. The inability to measure training data leakage implies that it remains unclear whether the observed performance in certain languages results from genuine crosslingual transfer or simply due to English data leakage during pretraining. This lack of clarity significantly impacts the interpretability of your results.
> >
> > >Moreover, in Figure 6, we conducted a controlled experiment where we explicitly finetuned the models on indomain content presented in English, where models still consistently perform better at answering questions in English than in other languages.
> >
> > This still invites some skepticism. The translation inconsistencies between versions of "Harry Potter" highlight that this dataset is not necessarily parallel, with notable variations in translations across languages. The superior performance in English might not indicate a crosslingual barrier but could instead suggest a skewed data selection process favoring languages more prevalent during pretraining. This distinction is critical, as it may undermine the experiment's conclusions about crosslingual transfer efficacy.
> >
> > My recommendation remains: consider removing the HP Quiz entirely and replacing it with content that has not been leaked on the internet, such as local company guidelines or university campus rules, as previously suggested in Q7. Re-running your experiments with such data would likely lead to a more robust experimental design, thereby enhancing the credibility of the results.
> >
> > Nevertheless, the expansion of the evaluation to include a broader set of multilingual models and languages marks a significant improvement in the study, and I am inclined to reassess my evaluation positively based on these revisions. However, I still believe the paper would benefit from further refinement, particularly in addressing the limitations discussed above. Strengthening the discussion around these points and acknowledging the inherent complexities of translation and pretraining data limitations would provide a more balanced and robust perspective.

---

> ### Author Response · Authors · 2024-12-02
>
> Thanks for the comments! Please find our response to your remaining questions.
>
> > While the general assertion that a multilingual human could transfer relevant knowledge is plausible, there are notable exceptions. Some translations differ so significantly that even speakers proficient in both languages may find it challenging to identify connections.
>
> Thank you for the valuable comment and for conducting the Russian translation experiment with native speakers. We agree that translating the domain-specific knowledge is a very challenging and complex task, as the reviewer highlighted. Our dataset’s focus on multiple-choice QA tasks might help mitigate the noise arising from translation variability by narrowing down the search space for the correct answer to a limited set of options.
>
> In addition to discussing the limitations of translation quality from Google Translate in Appendix A, we will clarify it in Section 3.2 when introducing the dataset. We will also recruit native speakers to perform translation quality checks in the revision.
>
> > The inability to measure training data leakage implies that it remains unclear whether the observed performance in certain languages results from genuine crosslingual transfer or simply due to English data leakage during pretraining. This lack of clarity significantly impacts the interpretability of your results.
>
> Thank you for your comment. We would like to highlight the value of the Harry Potter (HP) dataset in this context.
> - Given that the original books are written in English, the prevalence of HP content in English within the pretraining data is expected. Rather than being a concern, this provides an opportunity to study cross-lingual transfer for **off-the-shelf** models. Specifically, it allows us to examine whether models, when prompted in non-English languages, can link the question to their inherent knowledge (acquired predominantly in English) to answer questions effectively in other languages.
> - HP also represents a widely known and well-defined knowledge domain, with practical significance—for example, users may ask LLMs questions about HP in multiple languages to understand the book better. This makes it important and relevant to study this dataset.
>
> > Consider removing the HP Quiz entirely and replacing it with content that has not been leaked on the internet, such as local company guidelines or university campus rules, as previously suggested in Q7. Re-running your experiments with such data would likely lead to a more robust experimental design, thereby enhancing the credibility of the results
>
> We appreciate the reviewer’s suggestion to consider using local company guidelines or university campus rules, and we are committed to adding a new dataset like that in our revision. However, during our process of designing the curation of such datasets, we realized that if such content had not been included on the internet, it would not provide meaningful insights into the knowledge-intensive reasoning abilities of off-the-shelf models, as this type of content would not be included in their pretraining data. Instead, this content is more suited to fine-tuning experiments, where models can be trained on domain-specific datasets to adapt to such specialized knowledge.
>
> On the other hand, another important consideration is the extent to which such knowledge is truly domain-specific. Local company or school rules are less domain-specific compared to the fictional world of Harry Potter. These rules are often not unique to a particular organization; many of the rules may be general and shared across multiple institutions or companies across the world speaking different languages. As a result, even if we use such knowledge for fine-tuning, it may not provide clean insights for domain-specific knowledge-reasoning, and the models may have known such common rules in multiple languages.
>
> Our current focus on widely recognized Harry Potter content provides a benchmark for evaluating cross-lingual transfer and knowledge-intensive reasoning in off-the-shelf models and fine-tuned models. Therefore, we respectively disagree that the HP dataset is not appropriate. While we are happy to add additional dataset, we were unable to complete this during the rebuttal timeline. Nonetheless, we believe that additional datasets would only further support our observations and would not fundamentally alter our conclusions. The current results are both sufficient and significant as evidence of the cross-lingual knowledge barrier as shown in both general (MMLU) and domain-specific (HP) knowledge.
>
> We sincerely thank the reviewer for the time and insightful comments, which have helped us improve the manuscript.
> Please also let us know if there are other questions, and we look forward to the discussion with the reviewer to further enhance our work. Thank you!

---

### Official Review · Reviewer_xBQr · 2024-11-05

**Soundness:** 3
**Presentation:** 3
**Contribution:** 3
**Rating:** 8
**Confidence:** 4

**Summary:**

This paper investigates how multilingual LLMs perform for actual cross-lingual tasks. This covers a wide range of tasks including machine translation but also several variations of multilingual tasks where different parts of the input (text, multiple choice options, questions) are translated into different languages. The authors show a consistent drop in performance when considering actual mixed language tasks and suggest simple fine-tuning strategies to remedy this drop.

**Strengths:**

- well written paper
- interesting outcomes. Not entirely surprising but still nice to see.
- solid methodological evaluation (but see my comment regarding choice of language below)

**Weaknesses:**

- the languages considered are rather few and still relatively similar to each other: 2 germanic languages (English and German) and three romance languages (French, Italian, Spanish). In addition all languages are most likely very high resource (albeit not as high as English of course) during training of the different models. It would have been good to see languages that are rather different from the rest (Arabic or Chinese) maybe even low-resource (Bengali or Amharic).

- while multilingual have been used it would have been interesting to also consider models that do explicitly use cross-lingual supervision during training such as ALMA or Tower.

**Questions:**

In Footnote 1 you say "This criterion allows the case when a small amount of parallel texts accidentally crawled from the web are mixed in the pretraining dataset, as it is nearly impossible to verify. We believe such presence, if it exists at all, would have negligible impact on the model given the size of the rest of the pretraining data."

A) How do you know that GPT 3.5 and 4 don't use parallel data explicitly during training?

B) While the parallel data might be small in comparison to the rest of the data, does that really mean it has a negligible impact on the model, considering the vast amount of parameters that could allow for quite a high degree of memorisation? Can you substantiate your claim a bit better (references or otherwise)?

---

> ### Author Response · Authors · 2024-11-23
> **Response to Reviewer xBQr**
>
> Thank you for your feedback! We address your questions and comments below.
>
> > Q1: the languages considered are rather few and still relatively similar to each other: 2 germanic languages (English and German) and three romance languages (French, Italian, Spanish). In addition all languages are most likely very high resource (albeit not as high as English of course) during training of the different models. It would have been good to see languages that are rather different from the rest (Arabic or Chinese) maybe even low-resource (Bengali or Amharic).
>
> Thanks for the thoughtful comment.
> Following the suggestion, we have expanded our evaluation to include a total of 16 languages and 15 multilingual LLMs to provide broader insights into our study.
> Specifically, in addition to the 5 languages we originally evaluated, we further consider
> - *Low-resource languages*: Malay (ms), Danish (da), Finnish (fi), Norwegian (no), Bengali (bn), Amharic (am);
> - *Languages with token distributions significantly different from English*: Russian (ru), Chinese (zn), Hebrew (he), Arbic (ar) and Hindi (hi).
>
> Conclusions:
> -  The results on MMLU (Fig. 4 and Fig. 5) and the Harry Potter quiz (Fig. 7) show that cross-lingual knowledge barriers hold for those additional models and languages, which demonstrates the universality of our findings.
> - We also highlight that in Section 4.2, we evaluated our mixed-language fine-tuned LLMs on languages that were not used during the finetuning stage.  Our results in Figures 10 and 11  demonstrate that the model fine-tuned using our method on high-resource languages { English (en), French (fr), German (de), Spanish (es), and Italian (it)} can improve crosslingual performance on Harry potter quiz and MMLU for low-resource languages and languages that are rather different from English. These results provide  encouraging evidence for the generalizability of our fine-tuning approach.
>
> > Q2: while multilingual have been used it would have been interesting to also consider models that do explicitly use cross-lingual supervision during training such as ALMA or Tower.
>
> Thanks for the insightful comment. Following your suggestion, we evaluated a total of 15 multilingual LLMs, including the Tower 7B base model, the Tower 7B instruction-tuned model, and several state-of-the-art multilingual LLMs such as Qwen2.5-7B, Llama-3.1-8B, Mixtral-8x7B-v0.1, aya-expanse-8B, aya-23-8B, and Mistral-Nemo-Base-2407.
>
> The results, presented in MMLU (Fig. 4 and 5) and the Harry Potter Quiz (Fig. 7), indicate that even models explicitly trained with cross-lingual supervision, such as the Tower series, face similar cross-lingual knowledge barriers in both general-domain and domain-specific tasks. For example, in Fig. 4 of MMLU evaluation, these models exhibit a significant accuracy drop in mixed-language settings (e.g., question + GT-option, GT-option, and mixup translations) compared to monolingual settings (e.g., English-only or fully translated). This demonstrates that current LLMs struggle with understanding complex multilingual contexts and effectively linking parametric knowledge across languages to answer multiple-choice questions.
>
> Additionally, we observed that the Tower series LLMs, which are primarily optimized for translation-related tasks, have limited cross-lingual performance in knowledge-intensive tasks. This limitation could be attributed to factors such as catastrophic forgetting or constraints in the training data's knowledge coverage. Similarly, we envision that the ALMA series translation models would exhibit similar performance as the Tower series models.  These findings further highlight the contributions of our work in uncovering the weaknesses of existing multilingual LLMs and emphasizing the need for more robust cross-lingual understanding in future models.
>
>
> > Q3: In Footnote 1 you say "This criterion allows the case when a small amount of parallel texts accidentally crawled from the web are mixed in the pretraining dataset, as it is nearly impossible to verify. We believe such presence, if it exists at all, would have negligible impact on the model given the size of the rest of the pretraining data."
> A) How do you know that GPT 3.5 and 4 don't use parallel data explicitly during training?
> B) While the parallel data might be small in comparison to the rest of the data, does that really mean it has a negligible impact on the model, considering the vast amount of parameters that could allow for quite a high degree of memorisation? Can you substantiate your claim a bit better (references or otherwise)?
>
>
> Thanks for the thoughtful comment. We agree that verifying the presence or absence of parallel training materials in GPT-3.5/4 is challenging, and measuring their potential impact is equally difficult.  To address this uncertainty and avoid making unverifiable assumptions, we have revised the introduction section and removed the claim regarding the existence and impact of parallel data.

---

> > ### Comment · Reviewer_xBQr · 2024-11-27
> >
> > Thank you for your clarifications and the inclusion of the additional models. Given the limited time, this is a non-trivial effort and much appreciated.

---

> > > ### Author Response · Authors · 2024-11-27
> > > **Response to xBQr**
> > >
> > > Thank you again for your thoughtful comments and positive feedback. Your support is vital to us.

---

### Author Response · Authors · 2024-11-23
**Revision Summary**

We sincerely thank all reviewers for their constructive feedback and suggestions, which are very helpful to us.  We are encouraged that the reviewers found our work (1) provide interesting and useful insights on the cross-lingual ability of LLMs (Reviewer xBQr, 2t3Z, uRPF), (2) the results are solid and promising  (Reviewer xBQr, 2t3Z, uRPF) and (3) the writing is clear (Reviewer xBQr, 2t3Z).

Following the reviewers’ suggestions, we added more experiments/discussions, and we addressed the questions in the response to each reviewer. Below is a summary of our new experimental results in the revised PDF:

- **Figure 4, Figure 5, Figure 7**: We evaluated a total of 15 models on MMLU variants and the Harry Potter Quiz, including those explicitly stated to be multilingual, such as Qwen2.5-7B, Llama-3.1-8B, Mixtral-8x7B-v0.1, aya-expanse-8B, aya-23-8B, Mistral-Nemo-Base-2407, and the Tower series models. (Reviewer xBQr, 2t3Z)
- **Figure 5, Figure 7**: We expanded the evaluation to 16 languages on MMLU variants and the Harry Potter Quiz. These include low-resource languages such as Malay (ms), Danish (da), Finnish (fi), Norwegian (no), Bengali (bn), and Amharic (am), as well as languages with token distributions that differ significantly from English, such as Russian (ru), Chinese (zh), Hebrew (he), Arabic (ar), and Hindi (hi).  (Reviewer xBQr, 2t3Z)
- **Figures 11 and 12**: We evaluated mixed-language fine-tuned models on additional languages that were not seen during the fine-tuning stage, observing enhanced cross-lingual performance.  (Reviewer xBQr, 2t3Z)
- **Appendix Figures 14 and 15**: We provided detailed monolingual and cross-lingual evaluation results for each domain of MMLU, including STEM, humanities, social sciences, and others.

Please also let us know if there are other questions, and we look forward to the discussion with the reviewers to further improve our paper. Thank you!

---

### Meta-Review · Area_Chair_RsRa · 2024-12-19

**Metareview:**

The paper investigates the cross-lingual knowledge barrier for large language models (LLMs) by evaluating their performance on mixed-language tasks, general knowledge benchmarks (MMLU), and domain-specific evaluations (Harry Potter quiz). The paper observes that while LLMs demonstrate surface-level cross-lingual capabilities, they struggle with deeper cross-lingual transfer and propose fine-tuning on mixed-language datasets as a mitigation strategy. While the study raises an important topic of cross-lingual knowledge transfer, there are significant concerns regarding the methodology, experimental design, and overall contribution.

The strengths of the paper include its attempt to examine cross-lingual reasoning systematically and its clear writing. The expanded evaluation to include more languages and multilingual models is commendable. However, the weaknesses of the paper outweigh these strengths. First, the paper does not convincingly isolate the cross-lingual barrier phenomenon due to insufficiently controlled experiments and unclear disentanglement of confounding factors such as model multilinguality, token overlap, and pretraining data leakage. Many of the selected models, such as the Llama series, are not explicitly multilingual, raising concerns about whether their performance can be generalized. Second, the domain-specific evaluation setup using the Harry Potter dataset is problematic. Translation artifacts, inconsistent naming conventions, and the likelihood of pretraining data contamination undermine the reliability of the findings. Some reviewers argue that results from this dataset may reflect memorization rather than cross-lingual reasoning. Third, the proposed fine-tuning method to address the barrier lacks novelty. Techniques like mixed-language fine-tuning or code-switching have been previously explored, particularly for encoder-based models, and the improvements reported here are therefore incremental and expected. The paper also lacks discussion on the trade-offs of such fine-tuning. Finally, some aspects of the experimental presentation remains unclear.

Despite the authors' efforts to address these issues in the rebuttal, fundamental concerns about the experimental rigor and the novelty of the contribution persist. While the topic is relevant, the current version of the paper does not provide sufficiently strong evidence or a significant advancement in understanding or mitigating cross-lingual barriers in LLMs. For these reasons, I recommend rejection.

**Additional Comments On Reviewer Discussion:**

During the rebuttal period, the discussion primarily centred around the experimental design, dataset reliability, and the novelty of the contributions. The reviewers raised concerns regarding these aspects, and while the authors attempted to address the issues, significant doubts remained.

Experimental Design and Model Selection: Reviewers pointed out that several models evaluated, particularly the LLaMA series, are not explicitly multilingual, and their poor performance should not be generalized to multilingual LLMs. They also questioned the experimental controls, particularly whether the observed performance drop was due to a true "cross-lingual barrier" or model limitations. The authors expanded their evaluation to include 15 multilingual models and 16 languages, including low-resource ones, to strengthen their claims. However, concerns remained about the entanglement of confounding factors such as token overlap and pretraining biases.

Reliability of the Harry Potter Dataset: Reviewer 2t3Z criticized the domain-specific evaluation based on the Harry Potter quiz, highlighting translation inconsistencies, pretraining data contamination, and variability in name mappings across languages. Reviewer 2t3Z suggested replacing it with cleaner, non-leaked datasets. While the authors defended the dataset as useful for studying knowledge transfer and provided additional justification, they did not introduce alternative evaluations, citing time constraints.

Novelty of the Proposed Mitigation Methods: Reviewer uRPF argued that the fine-tuning methods proposed—such as mixed-language fine-tuning—are well-established, especially in encoder-only models, and lack novelty. The authors clarified that their work highlights the generalizability of such methods to decoder-based LLMs, but reviewers remained unconvinced that the improvements represented a significant advancement.

Presentation and Methodological Clarity: Reviewer uRPF highlighted ambiguities in embedding calculations, hyperparameter choices, and claims about model capabilities. The authors provided clarifications in the rebuttal and revised the paper accordingly, but concerns about the rigour and clarity of the presentation persisted.

In weighing these points, the expanded evaluation was noted as a positive effort, but it did not fully resolve the fundamental issues. The Harry Potter dataset's reliability remained a significant concern, as it introduced noise and potential data contamination that undermined the conclusions. Additionally, the lack of experimental controls and methodological clarity made it difficult to isolate the cross-lingual barrier phenomenon convincingly. The novelty of the contributions was deemed insufficient, as the methods were incremental and largely expected.
While reviewers 2t3Z and xBQr acknowledged some improvements and raised their scores slightly, uRPF maintained that the paper lacked rigor and substantive contribution. Given the persistent weaknesses in experimental design, dataset reliability, and the novelty of the proposed methods, these points collectively outweighed the paper's strengths. As such, I leaned toward the reviewers' concerns and recommend rejection.

---

### Decision · Program_Chairs · 2025-01-22

Reject